# SPECTRAL LEARNING OF SHARED DYNAMICS BETWEEN GENERALIZED-LINEAR PROCESSES

## ABSTRACT

Across various science and engineering applications, there often arises a need to predict the dynamics of one data stream from another. Further, these data streams may have different statistical properties. Studying the dynamical relationship between such processes, especially for the purpose of predicting one from the other, requires accounting for their distinct statistics while also dissociating their shared dynamical subspace. Existing analytical modeling approaches, however, do not address both of these needs. Here we propose a path forward by deriving a novel analytical multi-step subspace identification algorithm that can learn a model for a primary generalized-linear process (called "predictor"), while also dissociating the dynamics shared with a secondary process. We demonstrate a specific application of our approach for modeling discrete Poisson point-process activity, while finding the dynamics shared with continuous Gaussian processes. In simulations, we show that our algorithm accurately prioritizes identification of shared dynamics. Further, we also demonstrate that the method can additionally model the residual dynamics that exist only in the predictor Poisson data stream, if desired. Similarly, we apply our algorithm on a biological dataset to learn models of dynamics in Poisson neural population spiking streams that predict dynamics in movement streams. Compared with existing Poisson subspace identification methods, models learned with our method decoded movements better and with lower-dimensional latent states. Lastly, we discuss regimes in which our assumptions might not be met and provide recommendations and possible future directions of investigation.

## 1 INTRODUCTION

Modeling the shared dynamics between temporally-structured observations with different statistical properties is useful across multiple application domains, including neuroscience and biomedical engineering (D'mello & Kory, 2015; Lu et al., 2021). However, building models of the dynamic relation between such signals is challenging for two key reasons. First, continuous- and discrete-valued observations exhibit different statistics, which the modeling approach must appropriately reconcile. Second, residual (i.e., unshared or unique) dynamics present in each observation stream can obscure and confound modeling of their shared dynamics (Allen et al., 2019; Stringer et al., 2019; Sani et al., 2021). Thus, the modeling approach also needs a way to accurately dissociate and prioritize identification of the shared dynamics. Current analytical methods do not simultaneously enable both of these capabilities, which is what we address here.

Linear dynamical state-space models (SSMs) are a commonly used framework for modeling dynamics using a low-dimensional latent variable that evolves over time (Paninski et al., 2010; Macke et al., 2015; Newman et al., 2023). Even though the past decade has seen an increased use of artificial neural networks and deep learning methods for training dynamical models of time-series data (Pandarinath et al., 2018; Hurwitz et al., 2021; Kramer et al., 2022; Schneider et al., 2023), analytical SSMs still remain widely popular due to their interpretability and broad applicability both in scientific investigations and in real-time engineering applications (Kao et al., 2015; Aghagolzadeh & Truccolo, 2016; Lu et al., 2021; Yang et al., 2021; Newman et al., 2023). For Gauss-Markov models with continuous Gaussian observations, subspace system identification (SSID) theory provides computationally efficient non-iterative algorithms for analytically learning state-space models, both with and without identification of shared dynamics and dissociation of intrinsic vs input-driven

activity(Van Overschee & De Moor, 1996; Katayama, 2005; Sani et al., 2021; Galgali et al., 2023; Vahidi et al., 2023). These methods, however, are not applicable to generalized-linear processes with non-Gaussian observations. While there has been work extending SSID to generalized-linear processes, such as Poisson and Bernoulli observations (Buesing et al., 2012; Stone et al., 2023), these methods only learn the dynamics of a single observation time-series rather than modeling shared dynamics between two time-series (see section 2.1). Finally, prior multimodal learning algorithms do not explicitly tease apart the shared vs. residual (disjoint) dynamics in a predictor (primary) time-series, but instead model the collective dynamics of two modalities in the same latent states (Abbaspourazad et al., 2021; Kramer et al., 2022; Ahmadipour et al., 2023).

Here we fill these methodological gaps by deriving a novel covariance-based SSID learning algorithm that (1) is applicable to generalized-linear processes, and (2) is capable, with its two-staged learning approach, of identifying with priority the shared dynamics between two processes before modeling residual (predictor-only) dynamics. To illustrate the method, we focus on the specific case of modeling Poisson-distributed discrete time-series while dissociating their shared dynamics with Gaussian-distributed continuous observations, which is of particular interest in neuroscience. However, we emphasize that our method can be extended to other output distributions in the generalized-linear model family (section 5). We show that our method successfully dissociated the shared dynamics between Poisson and Gaussian observations both in simulations and on a public non-human primate (NHP) dataset of discrete population spiking activity recorded during continuous arm movements (O'Doherty et al., 2017). Further, compared with existing Poisson SSID methods, our method more accurately decoded movements from Poisson spiking activity using lower-dimensional latent states. Lastly, we discuss limitations and propose potential solutions and future research directions.

## 2 BACKGROUND

Our method provides the new capability to dynamically model Poisson observations, while prioritizing identification of dynamics shared with Gaussian observations. We first review the existing SSID method for modeling Poisson observations, which serves as our baseline, as well as standard covariance-based SSID, to help with the exposition of our method in section 3.

### 2.1 SSID FOR POISSON LINEAR DYNAMICAL SYSTEMS (PLDSID)

A Poisson linear dynamical system (PLDS) model is defined as

$$\begin{cases} \mathbf{x}_{k+1} & = \boldsymbol{A}\mathbf{x}_k + \mathbf{w}_k \\ \mathbf{r}_k & = \boldsymbol{C}_{\mathbf{r}}\mathbf{x}_k + \boldsymbol{b} \\ \mathbf{y}_k \mid \mathbf{r}_k & \sim \mathrm{Poisson}(\exp(\mathbf{r}_k)) \end{cases} \tag{1}$$

where $\mathbf{x}_k \in \mathbb{R}^{n_x}$ is the latent state variable and $\mathbf{y}_k \in \mathbb{R}^{n_y}$ corresponds to discrete (e.g., neural spiking) observations which, conditioned on the latent process $\mathbf{r}_k$, is Poisson-distributed with a rate equal to the exponential of $\mathbf{r}_k$ (i.e., log-rate). Finally, $\mathcal{N}(\mathbf{w}_k; \mathbf{0}, \boldsymbol{Q})$ is state noise and $\boldsymbol{b}$ is a constant baseline log-rate. The PLDS model is commonly used for modeling Poisson process events, such as neural spiking activity (Smith & Brown, 2003; Truccolo et al., 2005; Lawhern et al., 2010; Buesing et al., 2012; Macke et al., 2015). Buesing et al. (2012) developed an SSID algorithm, termed PLDSID, to learn the PLDS model parameters $\Theta = (\boldsymbol{A}, \boldsymbol{C_r}, \boldsymbol{b}, \boldsymbol{Q})$ given training samples $\boldsymbol{y}_k$ and hyperparameter $n_x$ corresponding to the latent state dimensionality.

There exist standard covariance-based SSID algorithms (section 2.2) that can learn the parameters of a latent dynamical system given a future-past Hankel matrix, $\boldsymbol{H}$, constructed from the cross-covariances of the system's linear observations as (Van Overschee & De Moor, 1996; Katayama, 2005)

$$\boldsymbol{H} := \mathrm{Cov}(\mathbf{r}_f, \mathbf{r}_p) = \begin{bmatrix} \boldsymbol{\Lambda}_{\mathbf{r}_i} & \boldsymbol{\Lambda}_{\mathbf{r}_{i-1}} & \cdots & \boldsymbol{\Lambda}_{\mathbf{r}_1} \\ \boldsymbol{\Lambda}_{\mathbf{r}_{i+1}} & \boldsymbol{\Lambda}_{\mathbf{r}_i} & \cdots & \boldsymbol{\Lambda}_{\mathbf{r}_2} \\ \vdots & \vdots & \cdots & \vdots \\ \boldsymbol{\Lambda}_{\mathbf{r}_{2i-1}} & \boldsymbol{\Lambda}_{\mathbf{r}_{2i-2}} & \cdots & \boldsymbol{\Lambda}_{\mathbf{r}_i} \end{bmatrix}, \quad \mathbf{r}_f := \begin{bmatrix} \mathbf{r}_i \\ \vdots \\ \mathbf{r}_{2i-1} \end{bmatrix}, \mathbf{r}_p := \begin{bmatrix} \mathbf{r}_0 \\ \vdots \\ \mathbf{r}_{i-1} \end{bmatrix}, \quad (2)$$

where the integer $i$ denotes the user-specified maximum temporal lag (i.e., horizon) used to construct $\boldsymbol{H}$ and $\boldsymbol{\Lambda}_{\mathbf{r}_\tau} := \mathrm{Cov}(\mathbf{r}_{k+\tau}, \mathbf{r}_k)$ is the $\tau$-th lag cross-covariance for any timepoint $k$, under

time-stationary assumptions. Such covariance-based SSID algorithms, however, are not directly applicable to Poisson-distributed observations (section 2.3). This is because the log-rates $\mathbf{r}_k$ that are linearly related to the latent states in equation (1) are not observable in practice – rather, only a stochastic Poisson emission from them (i.e., $\mathbf{y}_k$) is observed. As a result, the second moments constituting $\boldsymbol{H}$ (i.e., $\boldsymbol{\Lambda}_{\mathbf{r}_\tau}$) cannot be directly estimated. The critical insight by Buesing et al. (2012) was to leverage the log link function (i.e., $\exp^{-1}$) and the known conditional distribution $\mathbf{y}_k|\mathbf{r}_k$ to compute the first ($\boldsymbol{\mu}_{\mathbf{r}\pm}$) and second ($\boldsymbol{\Lambda}_{\mathbf{r}\pm}$) moments of the log-rate $\mathbf{r}_k$ from the first ($\boldsymbol{\mu}_{\mathbf{y}\pm}$) and second ($\boldsymbol{\Lambda}_{\mathbf{y}\pm}$) moments of the discrete observations $\mathbf{y}_k$. The $\pm$ denotes that moments are computed for the future-past stacked vector of observations $\mathbf{r}^\pm := \begin{bmatrix} \mathbf{r}_f^T & \mathbf{r}_p^T \end{bmatrix}^T$ and $\mathbf{y}^\pm := \begin{bmatrix} \mathbf{y}_f^T & \mathbf{y}_p^T \end{bmatrix}^T$, where

$$\boldsymbol{\mu}_{\mathbf{r}\pm} := E[\mathbf{r}^\pm] \quad \boldsymbol{\mu}_{\mathbf{y}\pm} := E[\mathbf{y}^\pm] \quad \boldsymbol{\Lambda}_{\mathbf{r}\pm} := \mathrm{Cov}(\mathbf{r}^\pm, \mathbf{r}^\pm) \quad \boldsymbol{\Lambda}_{\mathbf{y}\pm} := \mathrm{Cov}(\mathbf{y}^\pm, \mathbf{y}^\pm).$$

To compute moments of the log-rate, Buesing et al. (2012) derived the following moment conversion

$$
\begin{aligned}
\boldsymbol{\mu}_{\mathbf{r}_m^\pm} &= 2\ln(\boldsymbol{\mu}_{\mathbf{y}_m^\pm}) - \frac{1}{2}\ln(\boldsymbol{\Lambda}_{\mathbf{y}_{mm}^\pm} + \boldsymbol{\mu}_{\mathbf{y}_m^\pm}^2 - \boldsymbol{\mu}_{\mathbf{y}_m^\pm}) \\
\boldsymbol{\Lambda}_{\mathbf{r}_{mm}^\pm} &= \ln(\boldsymbol{\Lambda}_{\mathbf{y}_{mm}^\pm} + \boldsymbol{\mu}_{\mathbf{y}_m^\pm}^2 - \boldsymbol{\mu}_{\mathbf{y}_m^\pm}) - \ln(\boldsymbol{\mu}_{\mathbf{y}_m^\pm}^2) \\
\boldsymbol{\Lambda}_{\mathbf{r}_{mn}^\pm} &= \ln(\boldsymbol{\Lambda}_{\mathbf{y}_{mn}^\pm} + \boldsymbol{\mu}_{\mathbf{y}_m^\pm}\boldsymbol{\mu}_{\mathbf{y}_n^\pm}) - \ln(\boldsymbol{\mu}_{\mathbf{y}_m^\pm}\boldsymbol{\mu}_{\mathbf{y}_n^\pm})
\end{aligned}
\tag{3}
$$

where $m \neq n$ correspond to different indices of the first and second moments of the future-past stacked observation vectors $\mathbf{r}^\pm$ and $\mathbf{y}^\pm$, and $n, m = 1, \cdots, Kn_y$ where $K$ is the total number of time points. With the first and second moments computed in the moment conversion above, the baseline log rate $\boldsymbol{b}$ parameter is read off the first $n_y$ rows of $\boldsymbol{\mu}_{\mathbf{r}\pm}$ and the Hankel matrix, $\boldsymbol{H}$, is constructed as per equation (2). From here, it is possible to proceed with the standard covariance-based SSID algorithm for Gauss-Markov models using $\boldsymbol{H}$, as outlined next.

## 2.2 Standard Covariance-based SSID

Given an $\boldsymbol{H}$ matrix, covariance-based SSID first decomposes $\boldsymbol{H}$ into a product of observability ($\boldsymbol{\Gamma}_\mathbf{r}$) and controllability ($\boldsymbol{\Delta}$) matrices as (Van Overschee & De Moor, 1996; Katayama, 2005)

$$
\boldsymbol{H} \overset{\text{SVD}}{=} \boldsymbol{\Gamma}_\mathbf{r}\boldsymbol{\Delta} = \begin{bmatrix} \boldsymbol{C}_\mathbf{r} \\ \boldsymbol{C}_\mathbf{r}\boldsymbol{A} \\ \vdots \\ \boldsymbol{C}_\mathbf{r}\boldsymbol{A}^{i-1} \end{bmatrix} \begin{bmatrix} \boldsymbol{A}^{i-1}\boldsymbol{G} & \cdots & \boldsymbol{A}\boldsymbol{G} & \boldsymbol{G} \end{bmatrix}
\tag{4}
$$

where $\boldsymbol{G} := \mathrm{Cov}(\mathbf{x}_{k+1}, \mathbf{r}_k)$. The factorization of $\boldsymbol{H}$ is done by computing a SVD of $\boldsymbol{H}$ and keeping the top $n_x$ singular values and corresponding singular vectors. Note that the rank of $\boldsymbol{H}$ must be at least $n_x$ in order to identify a model with a latent dimension of $n_x$. Thus, the user-specified horizon $i$ must satisfy $i \times n_y \geq n_x$. From the factors of $\boldsymbol{H}$, $\boldsymbol{C}_\mathbf{r}$ is read off as the first $n_y$ rows of $\boldsymbol{\Gamma}_\mathbf{y}$ and $\boldsymbol{A}$ is learned by solving $\overline{\boldsymbol{\Gamma}}_\mathbf{r} = \underline{\boldsymbol{\Gamma}}_\mathbf{r}\boldsymbol{A}$, where $\overline{\boldsymbol{\Gamma}}_\mathbf{r}$ and $\underline{\boldsymbol{\Gamma}}_\mathbf{r}$ denote $\boldsymbol{\Gamma}_\mathbf{r}$ from which the top or bottom $n_y$ rows have been removed, respectively. This optimization problem has the following closed-form least-squares solution $\boldsymbol{A} = \underline{\boldsymbol{\Gamma}}_\mathbf{r}^\dagger\overline{\boldsymbol{\Gamma}}_\mathbf{r}$, with $\dagger$ denoting the pseudo-inverse operation. Discussion regarding learning the state noise covariance model parameter $\boldsymbol{Q}$ is postponed to section 3.2.3 below.

## 2.3 Challenges of developing Covariance- vs Projection-Based SSID methods

At a high-level, there exist two common approaches for subspace identification (Van Overschee & De Moor, 1996; Katayama, 2005): (i) covariance-based methods (e.g., Buesing et al. (2012); Ahmadipour et al. (2023)) that aim to learn all model parameters based on the second-order statistics of the observations and *not* the observation time-series directly, and (ii) projection-based methods (e.g., Sani et al. (2021); Vahidi et al. (2023)) that make direct use of the observation time-series via linear projections. Projection-based methods are often used to model Gaussian time-series but are not applicable to Poisson observations, which instead require a covariance-based approach since the latent log-firing rates ($\mathbf{r}$ in equation (1)) are unobserved. To achieve our aim (i.e., modeling a generalized-linear process with prioritized identification of shared dynamics), we need to develop a novel covariance-based subspace identification algorithm, which presents the following challenges:

1. Covariance-based methods, including PLDSID (Buesing et al., 2012), do not guarantee a valid set of parameters that satisfy the positive semidefinite covariance sequence requirements (Van Overschee & De Moor, 1996). To address this challenge, we use the optimization approach outlined in section 3.2.3 to ensure validity of noise statistics and enable inference from the learned model.

2. We could not rely on time-series projections to isolate the residual predictor process dynamics at the beginning of the second stage (Sani et al., 2021). As a result, we derived a new least-squares problem for learning the components of the state transition matrix $A$ corresponding to the unique predictor process dynamics (section 3.2.2), without changing the shared components learned in the first stage (section 3.2.1). By doing so, prioritized learning of shared dynamics is preserved.

## 3 METHOD

### 3.1 MODELING SHARED DYNAMICS BETWEEN POISSON AND GAUSSIAN OBSERVATIONS

The PLDS model (equation (1)) is for modeling Poisson observations on their own rather than with Gaussian observations. To enable this capability, we write the following Poisson-Gaussian linear dynamical system model

$$\begin{cases} \mathbf{x}_{k+1} &= \boldsymbol{A}\mathbf{x}_k + \mathbf{w}_k \\ \mathbf{z}_k &= \boldsymbol{C}_{\mathbf{z}}\mathbf{x}_k + \boldsymbol{\epsilon}_k \\ \mathbf{r}_k &= \boldsymbol{C}_{\mathbf{r}}\mathbf{x}_k + \boldsymbol{b} \\ \mathbf{y}_k \mid \mathbf{r}_k &\sim \text{Poisson}(\exp(\mathbf{r}_k)) \end{cases} \tag{5}$$

where $\mathbf{z}_k \in \mathbb{R}^{n_z}$ represents continuous observations (e.g., arm movements), $\boldsymbol{\epsilon}_k$ represents their noise (either white, i.e., zero-mean temporally uncorrelated Gaussian noise, or colored, i.e., zero-mean temporally correlated Gaussian noise), and $\mathbf{y}_k \in \mathbb{R}^{n_y}$ represents the discrete observations (i.e., neural spiking). Further, we introduce a block structure to the system (Sani et al., 2021) that allows us to dissociate shared latents from those that drive the Poisson observations only. Specifically,

$$\boldsymbol{A} = \begin{bmatrix} \boldsymbol{A}_{11} & \boldsymbol{0} \\ \boldsymbol{A}_{21} & \boldsymbol{A}_{22} \end{bmatrix} \quad \boldsymbol{C}_{\mathbf{z}} = \begin{bmatrix} \boldsymbol{C}_{\mathbf{z}}^{(1)} & \boldsymbol{0} \end{bmatrix} \quad \boldsymbol{C}_{\mathbf{r}} = \begin{bmatrix} \boldsymbol{C}_{\mathbf{r}}^{(1)} & \boldsymbol{C}_{\mathbf{r}}^{(2)} \end{bmatrix} \quad \mathbf{x} = \begin{bmatrix} \mathbf{x}^{(1)} \\ \mathbf{x}^{(2)} \end{bmatrix} \tag{6}$$

where $\mathbf{x}_k^{(1)} \in \mathbb{R}^{n_1}$ corresponds to latent states that drive both $\mathbf{z}_k$ and $\mathbf{y}_k$, and $\mathbf{x}_k^{(2)} \in \mathbb{R}^{n_x - n_1}$ to states that only drive $\mathbf{y}_k$. The parameter $\boldsymbol{G}$ can also be written in block partition format such that

$$\boldsymbol{G} = E\left[\begin{bmatrix} \mathbf{x}_{k+1}^{(1)} \\ \mathbf{x}_{k+1}^{(2)} \end{bmatrix} \mathbf{r}_k^T\right] - E\left[\begin{bmatrix} \mathbf{x}_{k+1}^{(1)} \\ \mathbf{x}_{k+1}^{(2)} \end{bmatrix}\right] E[\mathbf{r}_k]^T = \begin{bmatrix} E[\mathbf{x}_{k+1}^{(1)} \mathbf{r}_k^T] \\ E[\mathbf{x}_{k+1}^{(2)} \mathbf{r}_k^T] \end{bmatrix} - \begin{bmatrix} E[\mathbf{x}_{k+1}^{(1)}] E[\mathbf{r}_k]^T \\ E[\mathbf{x}_{k+1}^{(2)}] E[\mathbf{r}_k]^T \end{bmatrix} = \begin{bmatrix} \boldsymbol{G}^{(1)} \\ \boldsymbol{G}^{(2)} \end{bmatrix}.$$

Our method, termed PG-LDS-ID (Poisson-Gaussian linear dynamical system identification), learns the model parameters, i.e., $\Theta' = (\boldsymbol{A}, \boldsymbol{C}_{\mathbf{z}}, \boldsymbol{C}_{\mathbf{r}}, \boldsymbol{b}, \boldsymbol{Q})$, given training samples $\boldsymbol{y}_k$ and $\boldsymbol{z}_k$ and hyperparameters $n_1$ and $n_2 = n_x - n_1$ denoting the shared and residual latent dimensionalities. Selection of appropriate values for hyperparameters can be done with cross-validation (see appendix A.3).

### 3.2 PG-LDS-ID

PG-LDS-ID uses a two-staged learning approach to model Poisson time-series while prioritizing identification of the dynamics shared with Gaussian observations. During stage 1, shared dynamics are learned using both observations. In stage 2, any residual dynamics in the predictor observations are optionally learned. This two-staged approach allows prioritized learning of shared dynamics in the sense that latent states will be dedicated to explaining non-shared predictor dynamics only if there are enough latent states to explain the shared dynamics (full derivation in appendix A.1). Note, predictor refers to the data stream whose modeling is of primary interest (the Poisson observations here) and that is used to predict the secondary data stream (the Gaussian observations). For example, Poisson observations are the predictor within the context of decoding continuous behaviors from discrete population spiking activity. The roles can be swapped without loss of generality and the designation is made clear in equation (8) below.

### 3.2.1 STAGE 1: SHARED DYNAMICS

In the first stage, our algorithm identifies the parameter set corresponding to the shared dynamical subspace, $(\boldsymbol{A}_{11}, \boldsymbol{C}_{\mathbf{r}}^{(1)}, \boldsymbol{C}_{\mathbf{z}}, \boldsymbol{b})$, given hyperparameter $n_1$ and using both Gaussian and Poisson observations, $\mathbf{z}_k$ and $\mathbf{y}_k$. To do this, we first compute a moment conversion to estimate joint moments of $\mathbf{z}_k$ and $\mathbf{r}_k$ from the joint moments of the observed signals $\mathbf{z}_k$ and $\mathbf{y}_k$ with the following equation derived using the conditional statistical properties of both observations (see appendix A.1.4)

$$\boldsymbol{\Lambda}_{\mathbf{z}_{f_m}\mathbf{r}_{p_n}} \quad = \quad \mathrm{Cov}(\mathbf{z}_{f_m}, \mathbf{y}_{p_n}) \;/\; \boldsymbol{\mu}_{\mathbf{y}_{p_n}}. \tag{7}$$

Next, we use these moments to construct a Hankel matrix between future continuous observations and past log-rates of the discrete observations

$$\boldsymbol{H}_{\mathbf{zr}} := \mathrm{Cov}(\mathbf{z}_f, \mathbf{r}_p) = \begin{bmatrix} \boldsymbol{\Lambda}_{\mathbf{zr}_i} & \boldsymbol{\Lambda}_{\mathbf{zr}_{i-1}} & \cdots & \boldsymbol{\Lambda}_{\mathbf{zr}_1} \\ \boldsymbol{\Lambda}_{\mathbf{zr}_{i+1}} & \boldsymbol{\Lambda}_{\mathbf{zr}_i} & \cdots & \boldsymbol{\Lambda}_{\mathbf{zr}_2} \\ \vdots & \vdots & \cdots & \vdots \\ \boldsymbol{\Lambda}_{\mathbf{zr}_{2i-1}} & \boldsymbol{\Lambda}_{\mathbf{zr}_{2i-2}} & \cdots & \boldsymbol{\Lambda}_{\mathbf{zr}_i} \end{bmatrix}, \quad \mathbf{z}_f := \begin{bmatrix} \mathbf{z}_i \\ \vdots \\ \mathbf{z}_{2i-1} \end{bmatrix} \tag{8}$$

with $\mathbf{r}_p$ defined as in equation (2). Although equation (8) uses the same horizon for both observations, in practice we implement the method for a more general version with distinct horizon values $i_{\mathbf{y}}$ for the discrete observations and $i_{\mathbf{z}}$ for the continuous observations, resulting in $\boldsymbol{H}_{\mathbf{zr}} \in \mathbb{R}^{i_z * n_z \times i_{\mathbf{y}} * n_y}$. This allows users to independently specify the horizons for the two observations, which can improve modeling accuracy especially if the two observations have very different dimensionalities (see section 4.2 and appendix A.3.1). Further, and importantly, by using $\mathbf{z}$ as the future observations in the Hankel matrix, we learn a dynamical model wherein Poisson observations $\mathbf{y}$ can be used to predict the Gaussian observations $\mathbf{z}$. After constructing $\boldsymbol{H}_{\mathbf{zr}}$, we decompose it using SVD and keep the top $n_1$ singular values and their corresponding singular vectors

$$\boldsymbol{H}_{\mathbf{zr}} \overset{\mathsf{SVD}}{=} \boldsymbol{\Gamma}_{\mathbf{z}} \boldsymbol{\Delta}^{(1)} = \begin{bmatrix} \boldsymbol{C}_{\mathbf{z}} \\ \boldsymbol{C}_{\mathbf{z}} \boldsymbol{A}_{11} \\ \vdots \\ \boldsymbol{C}_{\mathbf{z}} \boldsymbol{A}_{11}^{i-1} \end{bmatrix} \begin{bmatrix} \boldsymbol{A}_{11}^{i-1} \boldsymbol{G}^{(1)} & \cdots & \boldsymbol{A}_{11} \boldsymbol{G}^{(1)} & \boldsymbol{G}^{(1)} \end{bmatrix} \tag{9}$$

where $n_1$ is the user-specified dimensionality of the shared latent states $\mathbf{x}_k^{(1)}$, $\boldsymbol{\Gamma}_{\mathbf{z}}$ denotes the observability matrix for the continuous observations, and $\boldsymbol{\Delta}^{(1)}$ denotes the controllability matrix associated with the shared latent states (defined as in equations (4) and (17) in the appendix). At this point, we extract $\boldsymbol{C}_{\mathbf{z}}$ by reading off the first $n_z$ rows of $\boldsymbol{\Gamma}_{\mathbf{z}}$. To extract $\boldsymbol{C}_{\mathbf{r}}^{(1)}$ we first form $\boldsymbol{H}$ per equation (2) and extract the observability matrix for $\mathbf{r}$ associated with the shared latent dynamics, $\boldsymbol{\Gamma}_{\mathbf{y}}^{(1)}$, by right multiplying $\boldsymbol{H}$ with the pseudoinverse of $\boldsymbol{\Delta}^{(1)}$

$$\boldsymbol{H}\boldsymbol{\Delta}^{(1)\dagger} = \boldsymbol{\Gamma}_{\mathbf{r}}^{(1)} = \begin{bmatrix} \boldsymbol{C}_{\mathbf{r}}^{(1)} \\ \boldsymbol{C}_{\mathbf{r}}^{(1)} \boldsymbol{A}_{11} \\ \vdots \\ \boldsymbol{C}_{\mathbf{r}}^{(1)} \boldsymbol{A}_{11}^{i-1} \end{bmatrix}.$$

We then read $\boldsymbol{C}_{\mathbf{r}}^{(1)}$ from the first $n_y$ lines of $\boldsymbol{\Gamma}_{\mathbf{r}}^{(1)}$ (defined as in equation (20) in the appendix). The baseline log rate $\boldsymbol{b}$ is read off the first $n_y$ rows of $\boldsymbol{\mu}_{\mathbf{r}\pm}$ computed in the moment conversion from equation (3). Lastly, to learn the shared dynamics summarized by the parameter $\boldsymbol{A}_{11}$, we solve the optimization problem $\underline{\boldsymbol{\Delta}}^{(1)} = \boldsymbol{A}_{11}\overline{\boldsymbol{\Delta}}^{(1)}$ where $\underline{\boldsymbol{\Delta}}^{(1)}$ and $\overline{\boldsymbol{\Delta}}^{(1)}$ denote $\boldsymbol{\Delta}^{(1)}$ from which $n_y$ columns have been removed from the right or left, respectively. The closed-form least-squares solution for this problem is $\boldsymbol{A}_{11} = \underline{\boldsymbol{\Delta}}^{(1)}(\overline{\boldsymbol{\Delta}}^{(1)})^{\dagger}$. This concludes the learning of the desired parameters $(\boldsymbol{A}_{11}, \boldsymbol{C}_{\mathbf{r}}^{(1)}, \boldsymbol{C}_{\mathbf{z}}, \boldsymbol{b})$, given hyperparameter $n_1$, in stage 1.

### 3.2.2 STAGE 2: RESIDUAL DYNAMICS

After learning the shared dynamics, our algorithm can learn the residual dynamics in the predictor observations that were not captured by $\mathbf{x}_k^{(1)}$. Specifically, we learn the remaining parameters from

equation (6): $([\boldsymbol{A}_{21} \quad \boldsymbol{A}_{22}], \boldsymbol{C}_{\mathbf{r}}^{(2)})$, with hyperparameter $n_2 = n_x - n_1$ determining the unshared latent dimensionality. To do so, we first compute a "residual" Hankel matrix, $\boldsymbol{H}^{(2)}$, using $\boldsymbol{\Gamma}_{\mathbf{r}}^{(1)}$ and $\boldsymbol{\Delta}^{(1)}$ from stage 1 and decompose it using SVD, keeping the first $n_2$ singular values and vectors

$$\boldsymbol{H}^{(2)} = \boldsymbol{H} - \boldsymbol{\Gamma}_{\mathbf{r}}^{(1)} \boldsymbol{\Delta}^{(1)} \overset{\mathsf{SVD}}{=} \boldsymbol{\Gamma}_{\mathbf{r}}^{(2)} \boldsymbol{\Delta}^{(2)}. \tag{10}$$

With $\boldsymbol{C}_{\mathbf{r}}^{(2)}$, which corresponds to the first $n_y$ rows of $\boldsymbol{\Gamma}_{\mathbf{r}}^{(2)}$, we construct $\boldsymbol{C}_r = \begin{bmatrix} \boldsymbol{C}_r^{(1)} & \boldsymbol{C}_{\mathbf{r}}^{(2)} \end{bmatrix}$. We then use $\boldsymbol{\Delta}^{(2)}$ to form the controllability matrix $\boldsymbol{\Delta}$ as the concatenation of $\boldsymbol{\Delta}^{(1)}$ and $\boldsymbol{\Delta}^{(2)}$ (derivation in appendix A.1): $\boldsymbol{\Delta} = \begin{bmatrix} \boldsymbol{A}^{i-1}\boldsymbol{G} & \cdots & \boldsymbol{A}\boldsymbol{G} & \boldsymbol{G} \end{bmatrix} = \begin{bmatrix} \boldsymbol{\Delta}^{(1)} \\ \boldsymbol{\Delta}^{(2)} \end{bmatrix}$. Given $\boldsymbol{\Delta}$, we next extract $[\boldsymbol{A}_{21} \quad \boldsymbol{A}_{22}]$ by solving the problem $\underline{\boldsymbol{\Delta}}^{(2)} = [\boldsymbol{A}_{21} \quad \boldsymbol{A}_{22}]\overline{\boldsymbol{\Delta}}$ where

$$\underline{\boldsymbol{\Delta}}^{(2)} := \begin{bmatrix} [\boldsymbol{A}_{21} & \boldsymbol{A}_{22}]\boldsymbol{A}^{i-2}\boldsymbol{G} & \cdots & [\boldsymbol{A}_{21} & \boldsymbol{A}_{22}]\boldsymbol{G} \end{bmatrix}, \quad \overline{\boldsymbol{\Delta}} := \begin{bmatrix} \boldsymbol{A}^{i-2}\boldsymbol{G} & \cdots & \boldsymbol{G} \end{bmatrix}.$$

Concatenating all the sub-blocks together $\boldsymbol{A} = \begin{bmatrix} \boldsymbol{A}_{11} & \mathbf{0} \\ \boldsymbol{A}_{21} & \boldsymbol{A}_{22} \end{bmatrix}$, we now have all model parameters, $(\boldsymbol{A}, \boldsymbol{C}_{\mathbf{r}}, \boldsymbol{C}_{\mathbf{z}}, \boldsymbol{b})$, given hyperparameters $n_1$ and $n_2$, except state noise covariance $\boldsymbol{Q}$.

### 3.2.3 Noise statistics

Standard SSID algorithms (e.g., section 2.2) learn linear SSMs of the following form

$$\begin{cases} \mathbf{x}_{k+1} &= \boldsymbol{A}\mathbf{x}_k + \mathbf{w}_k \\ \mathbf{r}_k &= \boldsymbol{C}_{\mathbf{r}}\mathbf{x}_k + \mathbf{v}_k \end{cases} \tag{11}$$

where the new term $\mathcal{N}(\mathbf{v}_k; \mathbf{0}, \boldsymbol{R})$ corresponds to observation noise. State noise, $\mathbf{w}_k$, and observation noise, $\mathbf{v}_k$, can have a non-zero instantaneous cross-covariance $\boldsymbol{S} = \mathrm{Cov}(\mathbf{w}_k, \mathbf{v}_k)$. SSID in general does not assume any restrictions on the noise statistics. However, the Poisson observation model (equations (1) and (5)) has no additive Gaussian noise for $\mathbf{r}_k$ and instead exhibits Poisson noise in $\mathbf{y}_k$, when conditioned on $\mathbf{r}_k$. This means that $\mathbf{v}_k = \mathbf{0}$ in equation (5), and thus $\boldsymbol{R} = \mathbf{0}$ and $\boldsymbol{S} = \mathbf{0}$. Imposing these constraints is important for accurate parameter identification for Poisson observations, but was not previously addressed by Buesing et al. (2012). Thus, we require our algorithm to find a complete parameter set $\Theta'$ that is close to the learned $(\boldsymbol{A}, \boldsymbol{C}_{\mathbf{r}}, \boldsymbol{C}_{\mathbf{z}}, \boldsymbol{b})$ from the two stages in sections 3.2.1 and 3.2.2 *and* imposes the noise statistic constraints $\boldsymbol{R} = \mathbf{0}$ and $\boldsymbol{S} = \mathbf{0}$. To do this, inspired by Ahmadipour et al. (2023), we form and solve the following convex optimization problem to satisfy the noise statistics requirements

$$\underset{\boldsymbol{\Lambda}_{\mathbf{x}}}{\text{minimize}} \quad \|\boldsymbol{S}(\boldsymbol{\Lambda}_{\mathbf{x}})\|_F^2 + \|\boldsymbol{R}(\boldsymbol{\Lambda}_{\mathbf{x}})\|_F^2 \quad \text{such that } \boldsymbol{\Lambda}_{\mathbf{x}} \succeq 0, \; \boldsymbol{Q}(\boldsymbol{\Lambda}_{\mathbf{x}}) \succeq 0, \; \boldsymbol{R}(\boldsymbol{\Lambda}_{\mathbf{x}}) \succeq 0 \tag{12}$$

where $\boldsymbol{\Lambda}_{\mathbf{x}} := \mathrm{Cov}(\mathbf{x}_k, \mathbf{x}_k)$ denotes the latent state covariance and the following covariance relationships, derived from equation (11) (Van Overschee & De Moor, 1996), hold

$$\begin{cases} \boldsymbol{Q}(\boldsymbol{\Lambda}_{\mathbf{x}}) &= \boldsymbol{\Lambda}_{\mathbf{x}} & - & \boldsymbol{A}\boldsymbol{\Lambda}_{\mathbf{x}}\boldsymbol{A}^T \\ \boldsymbol{R}(\boldsymbol{\Lambda}_{\mathbf{x}}) &= \boldsymbol{\Lambda}_{\mathbf{r}_0} & - & \boldsymbol{C}_{\mathbf{r}}\boldsymbol{\Lambda}_{\mathbf{x}}\boldsymbol{C}_{\mathbf{r}}^T \\ \boldsymbol{S}(\boldsymbol{\Lambda}_{\mathbf{x}}) &= \boldsymbol{G} & - & \boldsymbol{A}\boldsymbol{\Lambda}_{\mathbf{x}}\boldsymbol{C}_{\mathbf{r}}^T. \end{cases} \tag{13}$$

This approach has multiple benefits. First, it finds noise statistics that are consistent with the assumptions of the model (e.g., $\boldsymbol{R} = \mathbf{0}$). Second, it enforces the validity of learned parameters, i.e., parameters corresponding to a valid positive semidefinite covariance sequence (see section 4.3). It also enables state prediction (see appendix A.4). Combining the previously found parameters and the matrix $\boldsymbol{Q}$ that corresponds to the minimizing solution $\boldsymbol{\Lambda}_{\mathbf{x}}$ of equation (12), we have the full parameter set $\Theta' = (\boldsymbol{A}, \boldsymbol{C}_{\mathbf{r}}, \boldsymbol{C}_{\mathbf{z}}, \boldsymbol{b}, \boldsymbol{Q})$. We used Python's CVXPY package to solve the semidefinite programming problem defined in equation (12) (Diamond & Boyd, 2016; Agrawal et al., 2018). For all of our comparisons against baseline, we learned the noise statistics associated with PLDSID's identified parameters using this approach, keeping the rest of the algorithm the same.

## 4 Experimental Results

### 4.1 Shared dynamics are accurately identified in simulations

We simulated Poisson and Gaussian observations from random models as per equation (5) to evaluate how well our method identified the shared dynamics between the two observations. All state

and observation dimensions were randomly selected and the corresponding system parameters generated to simulate stable and slow-decaying dynamics (see appendix A.5.1). We computed two performance metrics: 1) the normalized eigenvalue error between ground truth and identified shared dynamical modes (i.e, the eigenvalues of $A_{11}$ in equation (6)), and 2) the predictive power of the model when using discrete Poisson observations to predict continuous Gaussian observations in a held-out test set. This second metric allowed us to test our hypothesis that PG-LDS-ID's explicit modeling of the shared subspace improved decoding of Gaussian observations from Poisson observations compared with PLDSID (Buesing et al., 2012). To compute the first metric for PLDSID, which does not explicitly model shared dynamics, we needed to select the $n_1$ modes identified from the Poisson time-series *only* that were the most representative of the Gaussian time-series. To do so, we first trained PLDSID on Poisson observations and extracted the latent states. Then, we sorted these learned latent states based on their accuracy in predicting the Gaussian observations (appendix A.4). We computed the eigenvalues associated with the top $n_1$ most predictive latent states, which we considered as the shared modes identified by PLDSID. We computed the normalized eigenvalue error as $|\Psi_{\text{true}} - \Psi_{\text{id}}|_F / |\Psi_{\text{true}}|_F$, where $\Psi_{\text{true}}$ and $\Psi_{\text{id}}$ denote vectors containing the true and learned shared eigenvalues and $|\cdot|_F$ denotes the Frobenius norm.

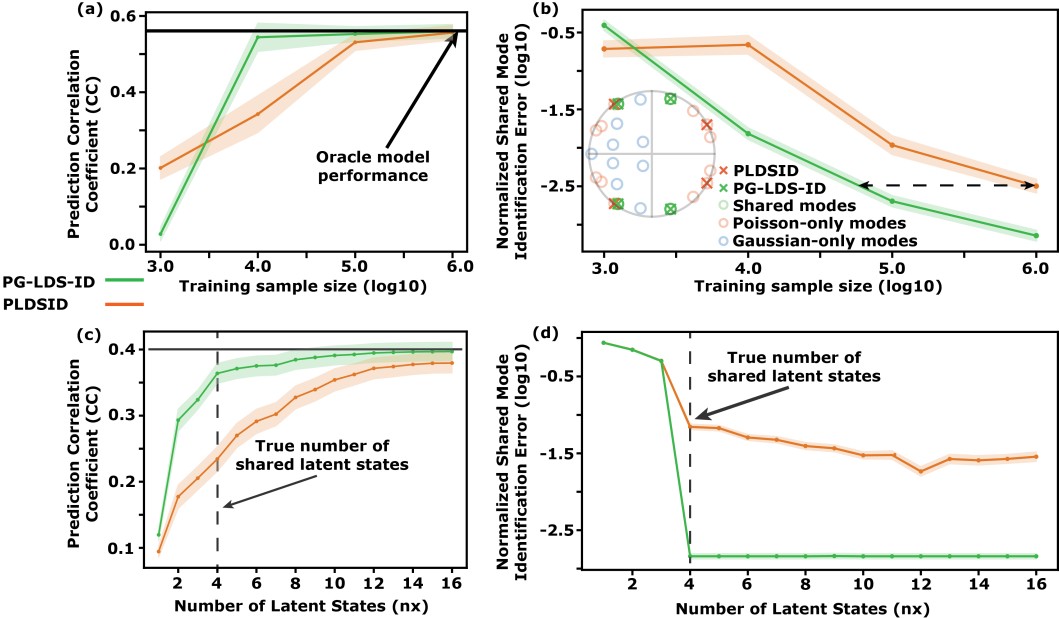

Figure 1: **In simulations, PG-LDS-ID more accurately learns the shared dynamical modes and better predicts the Gaussian observations from Poisson observations, especially in low-dimensional regimes**. Solid traces show the mean and the shaded areas denote the standard error of the mean (s.e.m.) for each condition. (a-b) Results for random models. Both the prediction correlation coefficient for the Gaussian observations in (a) and the normalized identification error of the shared dynamical modes (in log10 scale) in (b) are shown as a function of training samples used to learn the model parameters. (c-d) Same as (a-b) but for models with fixed shared ($n_1 = 4$) and residual ($n_2 = 12$) latent dimensions in the Poisson observations. PG-LDS-ID stage 1 used a dimensionality given by $\min(4, n_x)$. For configurations wherein learned $n_x$ is smaller than true $n_1$, we substituted missing modes with 0 prior to computing the normalized error.

In our first simulation experiment, we generated 50 random systems and studied the effect of training set size on learning. We used 1e2, 1e3, 1e4, 1e5 or 1e6 samples to train models and tested them on 1e6 samples of independent held-out data (figure 1). We found that our method required substantially fewer training samples (∼1e4 samples compared to PLDSID's ∼1e5) to reach ideal (i.e., ground truth) prediction (figure 1a). Similarly, our method more accurately identified the shared dynamical modes compared to PLDSID even when methods had increasingly more training samples (figure 1b). In our second simulation experiment, we studied the effect of latent state dimension on learning. We generated 50 systems with fixed dimensions for shared and total latent states given by

$n_1 = 4$ and $n_x = 16$, respectively. We swept the learned latent state dimension from 1 to the true dimensionality of $n_x = 16$, with the dimensionality of shared dynamics set to $\min(\text{current } n_x, n_1)$. We found that our method identified the correct shared modes with very small errors using only 4 latent state dimensions; in contrast, PLDSID did not reach such low error rates even when using higher latent state dimensions (figure 1d). In terms of predictive power, our method achieved close-to peak performance even when using as few as 4 latent states whereas PLDSID required much larger latent states dimensions, of around 16, to do so (figure 1c). Taken together, these results show the power of PG-LDS-ID for performing dimensionality reduction on Poisson observations while prioritizing identification of shared dynamics with a secondary Gaussian data stream.

## 4.2 Modeling shared and residual dynamics in Poisson population neural spiking activity improves motor decoding

As a demonstration on real data, we used our algorithm to model the shared dynamics between discrete population neural spiking activity and continuous arm movements in a publicly available NHP dataset from the Sabes lab (O'Doherty et al., 2017). The dataset is of a NHP moving a 2D-cursor in a virtual reality environment based on fingertip position. We use the 2D cursor position and velocity as the continuous observations $\mathbf{z}$. We removed channels that had average firing rates less than 0.5 Hz or greater than 100 Hz. Similar to Lawlor et al. (2018), we also removed channels that were correlated with other channels using a correlation coefficient threshold of 0.4. For all methods we used 50ms binned multi-unit spike counts for the discrete observations $\mathbf{y}$. We evaluated decoding performance of learned models using five-fold cross validation across six recording sessions. We performed cross-validation using randomly-selected, non-overlapping subsets of 15 channels ($n_y = 15$) within each session. We used a nested inner cross-validation to select hyperparameters per fold based on the prediction CC of kinematics in the training data. Hyperparameters in this context were discrete horizon $i_{\mathbf{y}}$, continuous horizon $i_{\mathbf{z}}$, and time lag, which specifies how much the neural time-series should be lagged to time-align with the corresponding behavioral time-series (Moran & Schwartz, 1999; Shoham et al., 2005; Pandarinath et al., 2018). We swept $i_{\mathbf{y}}$ values of 5 and 10 time bins, $i_{\mathbf{z}}$ values of 10, 20, 22, 25, 28, and 30 time bins; and lag values of 0, 2, 5, 8, and 10 time bins. To train PG-LDS-ID, we use the shared dynamics dimensionality of $n_1 = \min(\text{current } n_x, 8)$. We chose a maximum $n_1$ of 8 because behavior decoding roughly plateaued at this dimension.

Compared with PLDSID, our method learned models that led to better behavioral decoding at all latent state dimensions (figure 2a) and achieved a higher behavior decoding at the maximum latent state dimension. This result suggests that our method better learns the shared dynamics between Poisson spiking and continuous movement observations due to its ability to dissociate shared vs. residual latent states in Poisson observations. Interestingly, despite the focus on learning the shared latent states in the first stage, PG-LDS-ID was also able to extract the residual latent states in Poisson observations because of its second stage. This led to PG-LDS-SID performing similarly to PLDSID in terms of peak neural self-prediction AUC while outperforming PLDSID in terms of peak behavior decoding (figure 2c). Indeed, even with the inclusion of just two additional latent states to model residual Poisson dynamics ($n_2 = 2$, $n_x = 10$), neural self-prediction was comparable to models learned by PLDSID (figure 2b). Taken together, our method was extensible to real data and helped boost decoding performance, especially in low-dimensional latent regimes, by better identifying shared dynamics between Poisson and Gaussian observations. In appendix A.8 we also include preliminary results comparing against PLDS models fit using EM on a subset of this dataset.

## 4.3 Limitations

PG-LDS-ID, similar to other SSID methods, uses a time-invariant model which may not be suitable if the data exhibits non-stationarity, e.g., in chronic neural recordings. In such cases one would need to intermittently refit the model or develop adaptive extensions (Ahmadipour et al., 2021). Moreover, as with other covariance-based SSID methods, our method may be sensitive to the accuracy of the empirical estimates of the first- and second-order moments. However, with increasing number of samples these empirical estimates will approach true statistical values, thereby improving overall performance, as seen in figure 1a-b. Further, it may be possible that SSID methods fail to learn a valid set of parameters corresponding to a positive-definite covariance sequence (Van Overschee & De Moor, 1996; Katayama, 2005) or they may learn unstable state dynamics, meaning $\boldsymbol{A}$ has some eigenvalues with magnitude greater than 1 (see appendix A.6). These issues can arise due to errors

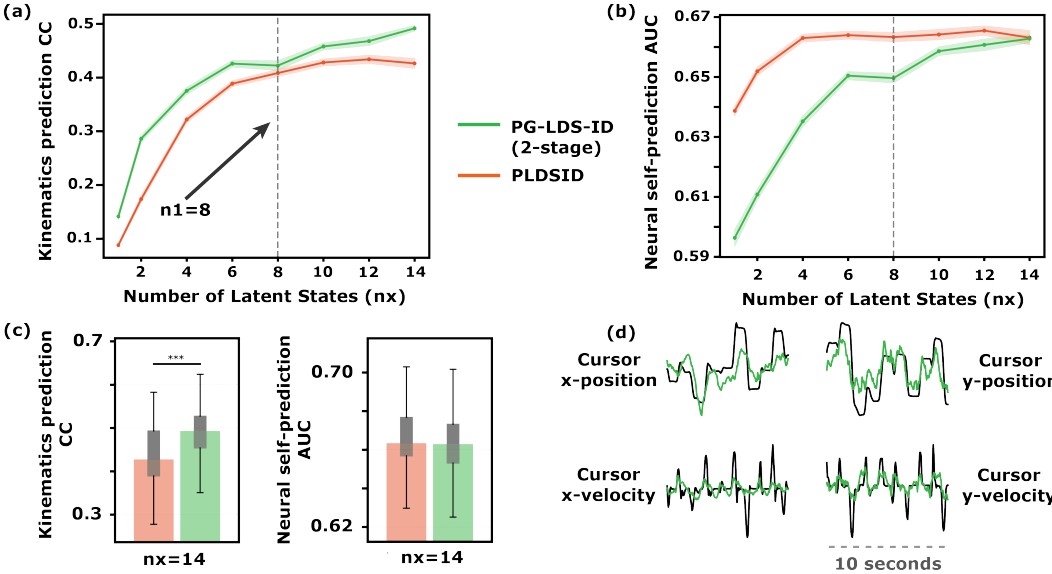

Figure 2: **In NHP data, PG-LDS-ID improves movement decoding from Poisson population spiking activity.** (a) Solid traces show the average cross-validated kinematic prediction CC and the shaded areas denote the s.e.m. for Poisson models of different latent dimensions learned by PG-LDS-ID (green) and PLDSID (orange). (b) Same as (a) but visualizing one-step ahead neural self-prediction AUC. (c) The left bar plots visualize the kinematic prediction CC and the right bar plots visualize the neural self-prediction AUC for models of latent dimensionality $n_x = 14$. We used Wilcoxon signed-rank test to measure significance. Asterisks in kinematic prediction CC plot indicate statistical significance with $p < 0.0005$; neural self-prediction AUCs were not significantly different at $n_x = 14$. (d) Example decoding of cursor (x,y) position and velocity from test data.

in the empirical estimates of the covariances and because these methods do not explicitly impose stability constraints on model parameters. Future work may consider incorporating techniques from control theory, such as mode stabilization and covariance matching, to help mitigate these limitations (Maciejowski, 1995; Lindquist & Picci, 1996; Byrnes et al., 1998; Alkire & Vandenberghe, 2002). Finally, our modeling approach can only provide an approximation of nonlinear dynamics/systems within the class of generalized-linear models, which have been shown to well-approximate nonlinear data in many applications, including modeling of neural and behavioral data.

## 5  DISCUSSION

We developed a novel analytical two-staged subspace identification algorithm termed PG-LDS-ID for modeling Poisson data streams while dissociating the dynamics shared with Gaussian data streams. Using simulations and real NHP data, we demonstrated that our method successfully achieves this new capability and thus, compared to existing Poisson SSID methods, more accurately identifies Poisson dynamics that are shared with Gaussian observations. Furthermore, this capability allows our method to improve decoding performance despite using lower-dimensional latent states and requiring a fewer number of training samples. Although we specifically focused on modeling Gaussian and Poisson observations, our algorithm can be extended to alternate distributions described with generalized-linear models. Our algorithm only requires the second-order moments after moment conversion (see equations (2), (3), (7), (8)). Because the moment conversion algorithm can be modified for the desired link function in generalized-linear models, as explained by Buesing et al. (2012), we can combine our method with the appropriate moment conversion to extend it to other non-Gaussian and non-Poisson observation distributions. Due to the high-prevalence of generalized-linear models across various application domains (e.g., biomedical engineering, neuroscience, finance, etc.), our method can be a general tool for modeling shared and residual dynamics of joint data streams with distinct observation distributions.

## 6 Reproducibility Statement

We have taken a few steps to ensure reproducibility of the results reported here. First, we are sharing the implementation of our algorithm, as supplementary material, along with example simulated data and a tutorial IPython notebook to demonstrate usage. Second, we used a publicly available dataset (O'Doherty et al., 2017) that can be easily accessed by anyone interested in reproducing the results reported in section 4.2. Finally, to further aid in reproducing results, we have also outlined the preprocessing and analyses steps we have taken in section 4.2 and appendix A.5.2.

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

# A APPENDIX

## A.1 DERIVATION

Here we provide the derivation for a prioritized covariance-based subspace identification algorithm that learns a dynamical model of a predictor data stream while dissociating shared vs. residual latents with a secondary data stream. We define the following equivalent formulation for our dynamical model (equation (5)), where the block structure delineates shared and unshared latent states

$$
\begin{cases}
\begin{bmatrix} \mathbf{x}_{k+1}^{(1)} \\ \mathbf{x}_{k+1}^{(2)} \end{bmatrix} &= \begin{bmatrix} \boldsymbol{A}_{11} & \boldsymbol{0} \\ \boldsymbol{A}_{21} & \boldsymbol{A}_{22} \end{bmatrix} \begin{bmatrix} \mathbf{x}_k^{(1)} \\ \mathbf{x}_k^{(2)} \end{bmatrix} + \mathbf{w}_k \\[2ex]
\mathbf{z}_k &= \begin{bmatrix} \boldsymbol{C}_{\mathbf{z}}^{(1)} & \boldsymbol{0} \end{bmatrix} \begin{bmatrix} \mathbf{x}_k^{(1)} \\ \mathbf{x}_k^{(2)} \end{bmatrix} + \boldsymbol{\epsilon}_k \\[2ex]
\mathbf{r}_k &= \begin{bmatrix} \boldsymbol{C}_{\mathbf{r}}^{(1)} & \boldsymbol{C}_{\mathbf{r}}^{(2)} \end{bmatrix} \begin{bmatrix} \mathbf{x}_k^{(1)} \\ \mathbf{x}_k^{(2)} \end{bmatrix} + \boldsymbol{b} \\[2ex]
\mathbf{y}_k \mid \mathbf{r}_k &\sim \mathrm{Poisson}(\exp(\mathbf{r}_k))
\end{cases}
\tag{14}
$$

with parameters and noise terms defined as in sections 2.1 and 3.1.

### A.1.1 STANDARD COVARIANCE-BASED SSID

Before we present the derivation for PG-LDS-ID, we review a few steps in standard covariance-based SSID (section 2.2) that will help us in the derivation. First, it can be shown that the $\tau$-th lag cross-covariance terms for $\mathbf{r}$ can be written in terms of model parameters as $\boldsymbol{\Lambda}_{\mathbf{r}_\tau} = \mathrm{Cov}\left(\mathbf{r}_{k+\tau}, \mathbf{r}_k\right) = \boldsymbol{C}_{\mathbf{r}} \boldsymbol{A}^{\tau-1} \boldsymbol{G}$, where $\boldsymbol{G} := \mathrm{Cov}(\mathbf{x}_{k+1}, \mathbf{r}_k)$. Using this relationship, the Hankel matrix, $\boldsymbol{H}$, can be expanded as (Van Overschee & De Moor, 1996; Katayama, 2005)

$$
\begin{aligned}
\boldsymbol{H} = \mathrm{Cov}(\mathbf{r}_f, \mathbf{r}_p) &= \begin{bmatrix} \boldsymbol{\Lambda}_{\mathbf{r}_i} & \boldsymbol{\Lambda}_{\mathbf{r}_{i-1}} & \cdots & \boldsymbol{\Lambda}_{\mathbf{r}_1} \\ \vdots & \vdots & \cdots & \vdots \\ \boldsymbol{\Lambda}_{\mathbf{r}_{2i-1}} & \boldsymbol{\Lambda}_{\mathbf{r}_{2i-2}} & \cdots & \boldsymbol{\Lambda}_{\mathbf{r}_i} \end{bmatrix} \\[2ex]
&= \begin{bmatrix} \boldsymbol{C}_{\mathbf{r}} \boldsymbol{A}^{i-1} \boldsymbol{G} & \boldsymbol{C}_{\mathbf{r}} \boldsymbol{A}^{i-2} \boldsymbol{G} & \cdots & \boldsymbol{C}_{\mathbf{r}} \boldsymbol{G} \\ \vdots & \vdots & \cdots & \vdots \\ \boldsymbol{C}_{\mathbf{r}} \boldsymbol{A}^{2i-2} \boldsymbol{G} & \boldsymbol{C}_{\mathbf{r}} \boldsymbol{A}^{2i-3} \boldsymbol{G} & \cdots & \boldsymbol{C}_{\mathbf{r}} \boldsymbol{A}^{i-1} \boldsymbol{G} \end{bmatrix}.
\end{aligned}
\tag{15}
$$

Second, using a singular-value decomposition, the above Hankel matrix $\boldsymbol{H}$ can be decomposed into observability, $\boldsymbol{\Gamma}_{\mathbf{r}}$, and controllability, $\boldsymbol{\Delta}$, matrices from which model parameters can be extracted (Van Overschee & De Moor, 1996; Katayama, 2005)

$$
\boldsymbol{H} \overset{\mathsf{SVD}}{=} \boldsymbol{\Gamma}_{\mathbf{r}} \boldsymbol{\Delta} = \begin{bmatrix} \boldsymbol{C}_{\mathbf{r}} \\ \boldsymbol{C}_{\mathbf{r}} \boldsymbol{A} \\ \vdots \\ \boldsymbol{C}_{\mathbf{r}} \boldsymbol{A}^{i-1} \end{bmatrix} \begin{bmatrix} \boldsymbol{A}^{i-1} \boldsymbol{G} & \cdots & \boldsymbol{A} \boldsymbol{G} & \boldsymbol{G} \end{bmatrix}.
\tag{16}
$$

### A.1.2 PRIORITIZED COVARIANCE-BASED SUBSPACE IDENTIFICATION: STAGE 1 DERIVATION

In the first stage of our algorithm, our goal is to learn the model parameters that correspond to the shared dynamical subspace of $\mathbf{z}$ and $\mathbf{r}$ via the latent state $\mathbf{x}_k^{(1)}$: $(\boldsymbol{A}_{11}, \boldsymbol{C}_{\mathbf{r}}^{(1)}, \boldsymbol{C}_{\mathbf{z}}, \boldsymbol{b})$. First, it can be shown that the $\tau$-th lag cross-covariance between $\mathbf{z}$ and $\mathbf{r}$ can be written in terms of model parameters as $\boldsymbol{\Lambda}_{\mathbf{zr}_\tau} = \mathrm{Cov}\left(\mathbf{z}_{k+\tau}, \mathbf{r}_k\right) = \boldsymbol{C}_{\mathbf{z}} \boldsymbol{A}^{\tau-1} \boldsymbol{G}$, where $\boldsymbol{G}$ is defined as before. Due to the block structure of equation (14), we have shown that $\boldsymbol{G}$ can be partitioned as $\boldsymbol{G} = \begin{bmatrix} \boldsymbol{G}^{(1)} \\ \boldsymbol{G}^{(2)} \end{bmatrix}$ (see

section 3.1). As a result, $\mathbf{\Delta}$ (equation (16)) also has a block-partition format

$$
\begin{aligned}
\mathbf{\Delta} &= \begin{bmatrix} \boldsymbol{A}^{i-1}\boldsymbol{G} & \cdots & \boldsymbol{A}\boldsymbol{G} & \boldsymbol{G} \end{bmatrix} \\
&= \begin{bmatrix} \begin{bmatrix} \boldsymbol{A}_{11} & \boldsymbol{0} \\ \boldsymbol{A}_{21} & \boldsymbol{A}_{22} \end{bmatrix} \boldsymbol{A}^{i-2}\boldsymbol{G} & \cdots & \begin{bmatrix} \boldsymbol{A}_{11} & \boldsymbol{0} \\ \boldsymbol{A}_{21} & \boldsymbol{A}_{22} \end{bmatrix} \begin{bmatrix} \boldsymbol{G}^{(1)} \\ \boldsymbol{G}^{(2)} \end{bmatrix} & \begin{bmatrix} \boldsymbol{G}^{(1)} \\ \boldsymbol{G}^{(2)} \end{bmatrix} \end{bmatrix} \\
&= \begin{bmatrix} \begin{bmatrix} \boldsymbol{A}_{11}^{i-1}\boldsymbol{G}^{(1)} \\ \begin{bmatrix} \boldsymbol{A}_{21} & \boldsymbol{A}_{22} \end{bmatrix} \boldsymbol{A}^{i-2}\boldsymbol{G} \end{bmatrix} & \cdots & \begin{bmatrix} \boldsymbol{A}_{11}\boldsymbol{G}^{(1)} \\ \begin{bmatrix} \boldsymbol{A}_{21} & \boldsymbol{A}_{22} \end{bmatrix} \boldsymbol{G} \end{bmatrix} & \begin{bmatrix} \boldsymbol{G}^{(1)} \\ \boldsymbol{G}^{(2)} \end{bmatrix} \end{bmatrix} = \begin{bmatrix} \mathbf{\Delta}^{(1)} \\ \mathbf{\Delta}^{(2)} \end{bmatrix}.
\end{aligned} \tag{17}
$$

and the cross-covariance term can be simplified as

$$
\mathbf{\Lambda}_{\mathbf{zr}_\tau} = \begin{bmatrix} \boldsymbol{C}_{\mathbf{z}}^{(1)} & \boldsymbol{0} \end{bmatrix} \begin{bmatrix} \boldsymbol{A}_{11} & \boldsymbol{0} \\ \boldsymbol{A}_{21} & \boldsymbol{A}_{22} \end{bmatrix}^{\tau-1} \begin{bmatrix} \boldsymbol{G}^{(1)} \\ \boldsymbol{G}^{(2)} \end{bmatrix} = \boldsymbol{C}_{\mathbf{z}}^{(1)} \boldsymbol{A}_{11}^{\tau-1} \boldsymbol{G}^{(1)}.
$$

Henceforth we drop the superscript $(1)$ on $\boldsymbol{C}_{\mathbf{z}}^{(1)}$, without loss of generality. The Hankel matrix between future continuous observations and past log-rates of the discrete observation can then be expanded as

$$
\begin{aligned}
\boldsymbol{H}_{\mathbf{zr}} = \mathrm{Cov}(\mathbf{z}_f, \mathbf{r}_p) &= \begin{bmatrix} \mathbf{\Lambda}_{\mathbf{zr}_i} & \mathbf{\Lambda}_{\mathbf{zr}_{i-1}} & \cdots & \mathbf{\Lambda}_{\mathbf{zr}_1} \\ \vdots & \vdots & \cdots & \vdots \\ \mathbf{\Lambda}_{\mathbf{zr}_{2i-1}} & \mathbf{\Lambda}_{\mathbf{zr}_{2i-2}} & \cdots & \mathbf{\Lambda}_{\mathbf{zr}_i} \end{bmatrix} \\
&= \begin{bmatrix} \boldsymbol{C}_{\mathbf{z}}\boldsymbol{A}_{11}^{i-1}\boldsymbol{G}^{(1)} & \boldsymbol{C}_{\mathbf{z}}\boldsymbol{A}_{11}^{i-2}\boldsymbol{G}^{(1)} & \cdots & \boldsymbol{C}_{\mathbf{z}}\boldsymbol{G}^{(1)} \\ \vdots & \vdots & \cdots & \vdots \\ \boldsymbol{C}_{\mathbf{z}}\boldsymbol{A}_{11}^{2i-2}\boldsymbol{G}^{(1)} & \boldsymbol{C}_{\mathbf{z}}\boldsymbol{A}^{2i-3}\boldsymbol{G}^{(1)} & \cdots & \boldsymbol{C}_{\mathbf{z}}\boldsymbol{A}_{11}^{i-1}\boldsymbol{G}^{(1)}. \end{bmatrix}.
\end{aligned}
$$

A singular-value decomposition of $\boldsymbol{H}_{\mathbf{zr}}$ yields the observability matrix for $\mathbf{z}$ (i.e., $\mathbf{\Gamma}_{\mathbf{z}}$) and the controllability matrix $\mathbf{\Delta}^{(1)}$ associated with the shared dynamics

$$
\boldsymbol{H}_{\mathbf{zr}} \overset{\mathsf{SVD}}{=} \mathbf{\Gamma}_{\mathbf{z}}\mathbf{\Delta}^{(1)} = \begin{bmatrix} \boldsymbol{C}_{\mathbf{z}} \\ \boldsymbol{C}_{\mathbf{z}}\boldsymbol{A}_{11} \\ \vdots \\ \boldsymbol{C}_{\mathbf{z}}\boldsymbol{A}_{11}^{i-1} \end{bmatrix} \begin{bmatrix} \boldsymbol{A}_{11}^{i-1}\boldsymbol{G}^{(1)} & \cdots & \boldsymbol{A}_{11}\boldsymbol{G}^{(1)} & \boldsymbol{G}^{(1)} \end{bmatrix}. \tag{18}
$$

At this point, $\boldsymbol{C}_{\mathbf{z}}$ can be read off the first $n_z$ rows of $\mathbf{\Gamma}_{\mathbf{z}}$. The shared latent dynamics matrix $\boldsymbol{A}_{11}$ can be learned by solving a least-squares problem based on the controllability matrix $\mathbf{\Delta}^{(1)}$ (as introduced in section 3.2.1)

$$
\underline{\mathbf{\Delta}}^{(1)} = \boldsymbol{A}_{11}\overline{\mathbf{\Delta}}^{(1)} \quad \text{where} \tag{19}
$$

$$
\underline{\mathbf{\Delta}}^{(1)} := \begin{bmatrix} \boldsymbol{A}_{11}^{i-1}\boldsymbol{G}^{(1)} & \cdots & \boldsymbol{A}_{11}\boldsymbol{G}^{(1)} \end{bmatrix}, \quad \overline{\mathbf{\Delta}}^{(1)} := \begin{bmatrix} \boldsymbol{A}_{11}^{i-2}\boldsymbol{G}^{(1)} & \cdots & \boldsymbol{G}^{(1)} \end{bmatrix},
$$

which has the following closed-form solution: $\boldsymbol{A}_{11} = \underline{\mathbf{\Delta}}^{(1)}(\overline{\mathbf{\Delta}}^{(1)})^\dagger$.

To extract $\boldsymbol{C}_{\mathbf{r}}$, we first note that the Hankel expansion in equation (16) can be simplified due to the block-structure of $\mathbf{\Delta}$ and $\mathbf{\Gamma}_{\mathbf{r}}$, which is defined as

$$
\mathbf{\Gamma}_{\mathbf{r}} = \begin{bmatrix} \boldsymbol{C}_{\mathbf{r}} \\ \boldsymbol{C}_{\mathbf{r}}\boldsymbol{A} \\ \vdots \\ \boldsymbol{C}_{\mathbf{r}}\boldsymbol{A}^{i-1} \end{bmatrix} = \begin{bmatrix} \boldsymbol{C}_r^{(1)} & \boldsymbol{C}_r^{(2)} \\ \boldsymbol{C}_r^{(1)}\boldsymbol{A} & \boldsymbol{C}_r^{(2)}\boldsymbol{A} \\ \vdots & \vdots \\ \boldsymbol{C}_r^{(1)}\boldsymbol{A}^{i-1} & \boldsymbol{C}_r^{(2)}\boldsymbol{A}^{i-1} \end{bmatrix} = \begin{bmatrix} \mathbf{\Gamma}_{\mathbf{r}}^{(1)} & \mathbf{\Gamma}_{\mathbf{r}}^{(1)} \end{bmatrix}, \tag{20}
$$

using the fact that $C_{\mathbf{r}} = \begin{bmatrix} C_{\mathbf{r}}^{(1)} & C_{\mathbf{r}}^{(2)} \end{bmatrix}$. Then, we explicitly separate $H = \mathbf{\Gamma_r}\mathbf{\Delta}$ (equation (16)) into two parts based on its singular-value decomposition as

$$
\begin{aligned}
H = \mathbf{U}\mathbf{\Sigma}\mathbf{V}^T \quad &= \overset{\mathbf{\Gamma_r}}{(\mathbf{U}\mathbf{\Sigma}^{1/2})} \overset{\mathbf{\Delta}}{(\mathbf{\Sigma}^{1/2}\mathbf{V}^T)} \\
&\overset{(a)}{=} \left( \begin{bmatrix} \mathbf{U}_{(1)} & \mathbf{U}_{(2)} \end{bmatrix} \begin{bmatrix} \mathbf{\Sigma}_{(1)}^{1/2} & \mathbf{0} \\ \mathbf{0} & \mathbf{\Sigma}_{(2)}^{1/2} \end{bmatrix} \right) \left( \begin{bmatrix} \mathbf{\Sigma}_{(1)}^{1/2} & \mathbf{0} \\ \mathbf{0} & \mathbf{\Sigma}_{(2)}^{1/2} \end{bmatrix} \begin{bmatrix} \mathbf{V}_{(1)}^T \\ \mathbf{V}_{(2)}^T \end{bmatrix} \right) \\
&= (\mathbf{U}_{(1)}\mathbf{\Sigma}_{(1)}^{1/2})(\mathbf{\Sigma}_{(1)}^{1/2}\mathbf{V}_{(1)}^T) + (\mathbf{U}_{(2)}\mathbf{\Sigma}_{(2)}^{1/2})(\mathbf{\Sigma}_{(2)}^{1/2}\mathbf{V}_{(2)}^T) \\
&= \mathbf{\Gamma_r}^{(1)}\mathbf{\Delta}^{(1)} + \mathbf{\Gamma_r}^{(2)}\mathbf{\Delta}^{(2)}
\end{aligned}
\tag{21}
$$

where simplification (a) is due to the block-partition structure of $\mathbf{\Gamma_r}$ and $\mathbf{\Delta}$. Thus $H$ can be written as the sum of "shared" and "residual" components (equation (21)). We can compute $\mathbf{\Gamma_r}^{(1)}$ as

$$
H\mathbf{\Delta}^{(1)\dagger} = (\mathbf{\Gamma_r}^{(1)}\mathbf{\Delta}^{(1)} + \mathbf{\Gamma_r}^{(2)}\mathbf{\Delta}^{(2)})\mathbf{\Delta}^{(1)\dagger} = \mathbf{\Gamma_r}^{(1)}
\tag{22}
$$

where we have used the orthonormal property of right singular vectors $\mathbf{V}$ to conclude $\mathbf{\Delta}^{(2)}\mathbf{\Delta}^{(1)\dagger} = \mathbf{0}$. At this point, we can extract $C_{\mathbf{r}}^{(1)}$ by reading the top $n_y$ rows of $\mathbf{\Gamma_r}^{(1)}$. Finally, $b$ is learned directly during the moment transformation (section 2.1). This concludes the learning of all parameters associated with the shared dynamical subspace, i.e., $(A_{11}, C_{\mathbf{r}}^{(1)}, C_{\mathbf{z}}, b)$.

### A.1.3 PRIORITIZED COVARIANCE-BASED SUBSPACE IDENTIFICATION: STAGE 2 DERIVATION

In the second stage of our algorithm, our goal is to learn model parameters that describe the residual dynamics of $\mathbf{r}$ via the latent state $\mathbf{x}_k^{(2)}$: $(\begin{bmatrix} A_{21} & A_{22} \end{bmatrix}, C_{\mathbf{r}}^{(2)})$. To learn these parameters, we first extract the residual component in equation (21), termed $H^{(2)}$, by subtracting $\mathbf{\Gamma_r}^{(1)}\mathbf{\Delta}^{(1)}$ from $H$, and decompose it via a singular-value decomposition to get $\mathbf{\Gamma_r}^{(2)}$ and $\mathbf{\Delta}^{(2)}$ as

$$
H^{(2)} = H - \mathbf{\Gamma_r}^{(1)}\mathbf{\Delta}^{(1)} \overset{\text{SVD}}{=} \mathbf{\Gamma_r}^{(2)}\mathbf{\Delta}^{(2)}.
\tag{23}
$$

At this point, we take $C_{\mathbf{r}}^{(2)}$ as the top $n_y$ rows of $\mathbf{\Gamma_r}^{(2)}$ and concatentate with $C_{\mathbf{r}}^{(1)}$ to complete $C_{\mathbf{r}}$.

To complete the state dynamics matrix $A$, we refer back to the block-structure representation of the controllability matrix in equation (17)

$$
\begin{bmatrix} \mathbf{\Delta}^{(1)} \\ \mathbf{\Delta}^{(2)} \end{bmatrix} = \left[ \begin{bmatrix} A_{11} & \mathbf{0} \\ A_{21} & A_{22} \end{bmatrix} A^{i-2}G \quad \cdots \quad \begin{bmatrix} A_{11} & \mathbf{0} \\ A_{21} & A_{22} \end{bmatrix} \begin{bmatrix} G^{(1)} \\ G^{(2)} \end{bmatrix} \quad \begin{bmatrix} G^{(1)} \\ G^{(2)} \end{bmatrix} \right]
$$

from which we construct the following relationship

$$
\begin{bmatrix} \underline{\mathbf{\Delta}}^{(1)} \\ \underline{\mathbf{\Delta}}^{(2)} \end{bmatrix} = \begin{bmatrix} A_{11} & \mathbf{0} \\ A_{21} & A_{22} \end{bmatrix} \begin{bmatrix} \overline{\mathbf{\Delta}}^{(1)} \\ \overline{\mathbf{\Delta}}^{(2)} \end{bmatrix}
\tag{24}
$$

where $\overline{\mathbf{\Delta}}$ and $\underline{\mathbf{\Delta}}$ are defined as in equation (19). We can further isolate the residual state transitions as the solution to the following equation (taken from the second row of equation (24))

$$
\underline{\mathbf{\Delta}}^{(2)} = \begin{bmatrix} A_{21} & A_{22} \end{bmatrix} \begin{bmatrix} \overline{\mathbf{\Delta}}^{(1)} \\ \overline{\mathbf{\Delta}}^{(2)} \end{bmatrix} = \begin{bmatrix} A_{21} & A_{22} \end{bmatrix} \overline{\mathbf{\Delta}},
\tag{25}
$$

which has the following closed-form least-squares solution: $\begin{bmatrix} A_{21} & A_{22} \end{bmatrix} = \underline{\mathbf{\Delta}}^{(2)}\overline{\mathbf{\Delta}}^{\dagger}$. The full state dynamics is the concatenation $A = \begin{bmatrix} A_{11} & \mathbf{0} \\ A_{21} & A_{22} \end{bmatrix}$. This concludes the learning of all parameters for the residual dynamics, i.e., $(\begin{bmatrix} A_{21} & A_{22} \end{bmatrix}, C_{\mathbf{r}}^{(2)})$.

### A.1.4 PG-LDS-ID: Transformation of joint Gaussian and Poisson moments

In sections A.1.1-A.1.3 we demonstrated how all model parameters can be extracted in two stages with prioritization, starting from the second-moments of the latent log-rates, $\mathbf{r}$, and the Gaussian observations, $\mathbf{z}$. Here we explain how we estimate these moments from the computable moments of $\mathbf{y}$ (the Poisson observations) and $\mathbf{z}$, using equation (7) as

$$\mathbf{\Lambda}_{\mathbf{z}_{f_m}\mathbf{r}_{p_n}} = \mathrm{Cov}(\mathbf{z}_{f_m}, \mathbf{y}_{p_n}) \,/\, \boldsymbol{\mu}_{\mathbf{y}_{p_n}}.$$

Here we provide a sketch of the proof. Without loss of generality, assume $\mathbf{z}$ and $\mathbf{r}$ are stationary with a mean of $\mathbf{0}$ (e.g., demeaned during preprocessing). We can compute the covariance of any two elements $j$ and $k$ of vectors $\mathbf{z}_f$ and $\mathbf{y}_p$ respectively as

$$\mathrm{Cov}(\mathbf{z}_{f_j}, \mathbf{y}_{p_k}) \quad = E\left[\mathbf{z}_{f_j}\mathbf{y}_{p_k}\right] = E\left[E\left[\mathbf{z}_{f_j}\mathbf{y}_{p_k}|\mathbf{r}_{p_k}\right]\right]$$

$$\stackrel{(a)}{=} E\left[E\left[\mathbf{z}_{f_j}|\mathbf{r}_{p_k}\right]\,E\left[\mathbf{y}_{f_k}|\mathbf{r}_{p_k}\right]\right] = E\left[E\left[\mathbf{z}_{f_j}|\mathbf{r}_{p_k}\right]\,\exp(\mathbf{r}_{p_k})\right]$$

where (a) is because $\mathbf{z}_{f_j}$ and $\mathbf{y}_{p_k}$ are independent when conditioned on latent log-rate $\mathbf{r}_{p_k}$. Next, we use the fact that $\mathbf{z}_f$ and $\mathbf{r}_p$ are jointly Gaussian random processes and, as a result, the mean of the conditional distribution, $E[\mathbf{z}_{f_j}|\mathbf{r}_{p_k}]$, is equal to $\Lambda_{\mathbf{z}_{f_j}\mathbf{r}_{p_k}}\Lambda^{-1}_{\mathbf{r}_{p_{kk}}}\mathbf{r}_{p_k}$ (i.e., the linear least-square estimate of $\mathbf{z}_{f_j}$ using $\mathbf{r}_{p_k}$). The last step is to compute the expectation

$$E\left[\Lambda_{\mathbf{z}_{f_j}\mathbf{r}_{p_k}}\Lambda^{-1}_{\mathbf{r}_{p_{kk}}}\mathbf{r}_{p_k}\,\exp(\mathbf{r}_{p_k})\right] = \mathbf{\Lambda}_{\mathbf{z}_{f_j}\mathbf{r}_{p_k}}\boldsymbol{\mu}_{\mathbf{y}_{p_k}} = \mathrm{Cov}(\mathbf{z}_{f_j}, \mathbf{y}_{p_k})$$

which, after rearranging terms, yields

$$\mathbf{\Lambda}_{\mathbf{z}_{f_m}\mathbf{r}_{p_n}} = \mathrm{Cov}(\mathbf{z}_{f_m}, \mathbf{y}_{p_n}) \,/\, \boldsymbol{\mu}_{\mathbf{y}_{p_n}}.$$

We note that the final equation is equivalent to a derivation provided by Buesing et al. (2012) as their supplementary equation (6) to compute cross-covariances between Poisson observations and Gaussian *inputs*, instead of between joint Poisson and Gaussian *observations* (as was in our case). The remaining unimodal (i.e., Poisson-only) moment conversions that are required to compute $\boldsymbol{H}$ are performed per equation (3) in section 2.1.

### A.1.5 Generalized cross-term Hankel matrix with different horizons per observation

For ease of exposition, the derivation in section A.1.2 was provided for a cross-term Hankel matrix $\boldsymbol{H}_{\mathbf{zr}}$ that was formed with equal horizons for $\mathbf{z}$ and $\mathbf{r}$ as

$$\boldsymbol{H}_{\mathbf{zr}} := \mathrm{Cov}(\mathbf{z}_f, \mathbf{r}_p) = \begin{bmatrix} \mathbf{\Lambda}_{\mathbf{zr}_i} & \mathbf{\Lambda}_{\mathbf{zr}_{i-1}} & \cdots & \mathbf{\Lambda}_{\mathbf{zr}_1} \\ \mathbf{\Lambda}_{\mathbf{zr}_{i+1}} & \mathbf{\Lambda}_{\mathbf{zr}_i} & \cdots & \mathbf{\Lambda}_{\mathbf{zr}_2} \\ \vdots & \vdots & \cdots & \vdots \\ \mathbf{\Lambda}_{\mathbf{zr}_{2i-1}} & \mathbf{\Lambda}_{\mathbf{zr}_{2i-2}} & \cdots & \mathbf{\Lambda}_{\mathbf{zr}_i} \end{bmatrix}, \quad \mathbf{z}_f := \begin{bmatrix} \mathbf{z}_i \\ \vdots \\ \mathbf{z}_{2i-1} \end{bmatrix}, \mathbf{r}_p := \begin{bmatrix} \mathbf{r}_0 \\ \vdots \\ \mathbf{r}_{i-1} \end{bmatrix}.$$

In general, the rank of Hankel matrices formed from ideal data covariances can be shown to be the same as the state dimension associated with it (Van Overschee & De Moor, 1996; Katayama, 2005), i.e., $n_1 = \mathrm{rank}(\boldsymbol{H}_{\mathbf{zr}})$ per equation (18) and $n_x = \mathrm{rank}(\boldsymbol{H})$ per equation (16). However, during system identification these Hankel matrices are formed from non-ideal empirical sample covariances and, as a result, are typically full rank. Nevertheless, we expect the singular values associated with real dynamics (e.g., the first $n_1$ singular values in $\boldsymbol{H}_{\mathbf{zr}}$) to be larger than subsequent singular values that are due to noise. Indeed, the goal of the SVD applied to Hankel matrices, e.g., in equations (16), (18), and (23), is to remove noisy singular values and only keep the largest singular values that are most likely due to real dynamics.

Given that the Hankel matrices formed during system identification are typically full rank, their rank is determined based on their dimensions, i.e., $\mathrm{rank}(\boldsymbol{H}_{\mathbf{zr}}) = \min(i \times n_y, i \times n_z)$ and $\mathrm{rank}(\boldsymbol{H}) = i \times n_y$. Thus, the horizon parameter $i$ that is used to form the Hankel matrix plays an important role in its final dimensions, rank, and, consequently, on the maximum number of non-zero singular values that can be preserved after applying SVD. This, in turn, determines the maximum state dimension that can be learned for the resulting model. Thus, to provide more flexibility over the state dimensions

that can be learned in each stage of PG-LDS-ID, we generalize the Hankel matrix $\boldsymbol{H_{zr}}$ to support different horizon values for each of the observations, $i_z$ and $i_y$, such that

$$
\boldsymbol{H_{zr}} = \begin{bmatrix} \boldsymbol{\Lambda_{zr}}_{i_z} & \boldsymbol{\Lambda_{zr}}_{i_z-1} & \cdots & \boldsymbol{\Lambda_{zr}}_{i_z-i_y+1} \\ \boldsymbol{\Lambda_{zr}}_{i_z+1} & \boldsymbol{\Lambda_{zr}}_{i_z} & \cdots & \boldsymbol{\Lambda_{zr}}_{i_z-i_y+2} \\ \vdots & \vdots & \cdots & \vdots \\ \boldsymbol{\Lambda_{zr}}_{2i_z-1} & \boldsymbol{\Lambda_{zr}}_{2i_z-2} & \cdots & \boldsymbol{\Lambda_{zr}}_{2i_z-i_y} \end{bmatrix} \quad \text{with} \quad \mathbf{z}_f := \begin{bmatrix} \mathbf{z}_{i_z} \\ \vdots \\ \mathbf{z}_{2i_z-1} \end{bmatrix}, \mathbf{r}_p := \begin{bmatrix} \mathbf{r}_0 \\ \vdots \\ \mathbf{r}_{i_y-1} \end{bmatrix}.
$$

The discrete observation horizon $i_y$ is also used when forming the Hankel matrix $\boldsymbol{H}$, per equation (15). The additional flexibility gained from having different horizon values is especially critical in scenarios wherein the dimensionalities of $\mathbf{z}$ and $\mathbf{y}$ are very different, such as in the case of our NHP analysis where $n_z = 4$ and $n_y = 15$. We select the final horizons $i_z$ and $i_y$ via an inner cross-validation based on which values achieve the best accuracy in the training data.

## A.2 GENERALIZABILITY OF THE BLOCK STRUCTURE FORMULATION

Here we explain how the blocked formulation in equation (6) can be assumed without loss of generality. The latent states in our model describe the primary data stream ($\mathbf{y}_k$, e.g., Poisson spiking activity), with a subset also explaining the secondary stream ($\mathbf{z}_k$, e.g., behavior). Formally, we define the true dimensionality of the shared states (denoted by $n_1$) based on the rank of the observability matrix for the pair $(\boldsymbol{A}, \boldsymbol{C_z})$. It can be shown using linear systems theory that an invertible linear transformation of the latent states (i.e., a similarity transformation) always exists that can place the $n_1$ dimensional latent subspace that is observable via $\mathbf{z}_k$ as the first few dimensions of the latent space, thus giving the block-structured formulation of equation (6). This can be seen by applying Theorem 3.8 from Katayama (2005) to the first two lines of equation (5). Thus, the blocked formulation of equation (6) is equivalent to the formulation from (5) and we can aim to learn our model in the form of equation (6) without any loss of generality. Moreover, note that this blocked formulation also covers the special case of a non-blocked formulation when $n_1 = n_x$, that is when all latent states contribute to both data streams and the observability matrix is full-rank. In this case, the top-left-block of $\boldsymbol{A}$ grows to cover the whole $\boldsymbol{A}$, and thus no zero-filled upper-right block would remain. The algorithm would still work in this special case by simply only applying the first stage of learning. However, within the application of modeling neural and behavioral data, we typically expect a minority of the neural dynamics to be related to a particular behavior of interest and so we expect the most appropriate $n_1$ to be smaller than $n_x$.

## A.3 SELECTION OF HYPERPARAMETERS

Hyperparameters $n_1$ and $n_x$ denote the number of shared vs. total latent state dimensions (equation (6)). When modeling real data, one can estimate the most appropriate values for these hyperparameters for the data using the following procedure:

1. Sweep over values of $n_1$, increasing $n_1$ while keeping $n_x = n_1$ (i.e., using stage 1 only to learn). Quantify the prediction of the secondary data stream (e.g., behavior) in each case to find the $n_1$ at which the prediction plateaus or reaches a peak. This value gives the appropriate $n_1$. Alternatively, $n_1$ can be estimated as the number of non-zero (or non-negligible) singular values of the Hankel matrix $\boldsymbol{H_{zr}}$ (equations (8) and (9)).

2. Using the selected $n_1$ from above, sweep over values of $n_x$, starting from $n_1$ and increasing the latent state dimension. Quantify the self-prediction of the predictor data stream (e.g., spiking activity) in each case, and find the $n_x$ at which the self-prediction reaches a peak.

### A.3.1 SECONDARY DATA STREAM NOISE STATISTICS AND CONSIDERATIONS

The model parameters stated in section 3.1 correspond to the parameters required by the point-process filter (Eden et al., 2004) for state estimation (see appendix A.4). However, if desired, under Gaussian assumptions the noise covariance term for the secondary data stream, $\boldsymbol{\Lambda_\epsilon}$, can be learned by computing the covariance of the prediction residuals as $E[(\hat{\mathbf{z}}_k - \mathbf{z}_k)(\hat{\mathbf{z}}_k - \mathbf{z}_k)^T]$, where $\hat{\mathbf{z}}_k$ denotes the predicted value of $\mathbf{z}$ and the expectation denotes the empirical average across all time samples. When $\epsilon_k$ is not white, the behavior prediction residuals under Gaussian assumptions can be computed in the same way (i.e., $\hat{\mathbf{z}}_k - \mathbf{z}_k$) and modeled using Gaussian SSID.

The Gaussian process' noise has an effect on learning similar to that of the residual dynamics present in the primary (i.e., Poisson) data stream: it reduces the overall signal-to-noise ratio of the shared dynamics present in both observation data streams. However, with increasingly more training samples and improved estimates of the second-order moments (i.e., covariances), the algorithm can become more robust to the impact of Gaussian observation noise.

## A.4  STATE PREDICTION

For model evaluations we chose to predict the continuous Gaussian observations from the discrete Poisson observations, which is a common use-case in neuroscience (Koyama et al., 2010; Macke et al., 2015; Lu et al., 2021). Once model parameters are learned in a training set, either with PLDSID or PG-LDS-ID, we can use the learned parameters to construct the Poisson point-process filter (PPF) (Eden et al., 2004) and estimate the latent states in a test set. Note, using the PPF for state estimation is only possible if noise statistics are valid (section 3.2.3). We denote the one-step ahead latent state prediction of $\mathbf{x}_k$ using all samples of $\mathbf{y}_k$ up to time $k-1$ by $\hat{\mathbf{x}}_{k|k-1}$. These state estimates can be used to predict the continuous observations as $C_{\mathbf{z}}\hat{\mathbf{x}}_{k|k-1}$. To learn a $C_{\mathbf{z}}$ parameter for PLDSID, we first estimate the latent states in the training data using a PPF and then fit a linear regression (scikit-learn) from the latent states to $\mathbf{z}_k$, i.e., $C_{\mathbf{z}} = \mathbf{Z}\hat{\mathbf{X}}^T(\hat{\mathbf{X}}\hat{\mathbf{X}}^T)^{\dagger}$, where columns of $\mathbf{Z}$ and $\hat{\mathbf{X}}$ contain $\mathbf{z}_k$ and $\hat{\mathbf{x}}_{k|k-1}$ for all training timepoints $k$ (Pedregosa et al., 2011). To make the methods more comparable, we use the same approach to refit the $C_z$ learned by PG-LDS-ID. We quantify the decoding performance using correlation coefficient (CC). We also assessed the one-step ahead self-prediction of Poisson observations using the predicted latent states. This was quantified with the area under the curve (AUC) of the receiver operating characteristic (appendix A.5.3).

## A.5  EXPERIMENTAL DETAILS

### A.5.1  SIMULATIONS

For our synthetic data in section 4.1, we simulated Poisson-Gaussian observations from random models as per equation (5). We randomly selected the latent state dimension $n_x$, the shared dimension $n_1$, and the observation dimensions $n_y$ and $n_z$ with uniform probability from the following ranges: $1 \leq n_x \leq 10$, $20 \leq n_y \leq 30$, $5 \leq n_z \leq 10$, and $1 \leq n_1 \leq n_x$. Using these dimensions we generated random model parameters $\Theta = (A, C_{\mathbf{r}}, C_{\mathbf{z}}, b, Q)$. We constrained the complex eigenvalues (i.e., modes) of the state transition matrix $A$ to have magnitudes uniformly distributed between $[0.93, 0.99]$ and phases uniformly distributed between $[0.019, 0.314]$. These restrictions correspond to stable, slow-decaying systems with time-constants within $[0.138, 0.995]$ seconds and frequencies within $[0.3, 5]$ Hz that are representative of various real time-series data, such as neural dynamics (Churchland et al., 2012; Song et al., 2022). All simulations were on a 10 ms timescale with a baseline log rate, $b$, randomly selected within $[0.5, 15]$ Hz. Observation matrix $C_{\mathbf{r}}$ was scaled to achieve a desired per-dimension maximum firing rate such that $\max_{1 \leq k \leq N} \exp(\mathbf{r}_k) \in [25, 65]$ Hz. This was to ensure a realistic range of firing rate and sufficient modulation depth for all dimensions of the simulated Poisson point-process. $Q$ was randomly generated to be a positive definite matrix and $C_{\mathbf{z}}$ was generated to hit a target signal-to-noise, defined as the variance associated with latent states normalized by observation noise variance $(C_{\mathbf{z}}\Lambda_{\mathbf{x}}C_{\mathbf{z}}^T)/(\Lambda_{\epsilon})$. Target SNR values were randomly generated as $10^{\alpha}$ with $\alpha$ uniformly distributed between $[0, 2]$. For every simulated model, the colored noise for the Gaussian process, $\epsilon_k$, was taken as the output of a 4-dimensional latent linear dynamical system with random parameters that were generated similarly. By using a general colored noise, we can simulate dynamics present in the continuous modality that are unshared with the discrete modality.

### A.5.2  NHP DATASETS

All NHP analyses were performed on a public dataset released by the Sabes lab (O'Doherty et al., 2017), using the following sessions from monkey I: 20160915/01, 20160916/01, 20160921/01, 20160927/04, 20160927/06, 20160930/02.

### A.5.3 NEURAL ONE-STEP AHEAD SELF-PREDICTION

To evaluate how well our algorithm modeled neural dynamics, we computed the one-step ahead self-prediction performance as quantified by AUC (section A.4). Our goal was to validate if our model could, when using all past neural observations, accurately predict the occurrence of spikes versus no spikes in a given time step. Since all self-predictions are made using the recursive point-process filter estimates of the latent states (see appendix A.4), we computed the probability of a spiking event for the $m$-th dimension of $\mathbf{y}$ at time $k$, conditioned on all observations $\mathbf{y}_{1:k-1}$, as

$$
\begin{aligned}
P(\mathbf{y}_k^m > 0 | \mathbf{y}_{1:k-1}) \quad &= \sum_{\mathbf{x}_k} p(\mathbf{y}_k^m > 0 \mid \mathbf{y}_{1:k-1}, \mathbf{x}_k) p(\mathbf{x}_k \mid \mathbf{y}_{1:k-1}) \\[6pt]
&\overset{(a)}{=} E_{\mathbf{x}_k \mid \mathbf{y}_{1:k-1}} \left[ p(\mathbf{y}_k^m > 0 \mid \mathbf{x}_k) \right] \overset{(b)}{=} E_{\mathbf{x}_k \mid \mathbf{y}_{1:k-1}} \left[ 1 - \exp(\exp(\mathbf{r}_k^m)) \mid \mathbf{x}_k \right] \\[6pt]
&\overset{(c)}{\approx} E_{\mathbf{x}_k \mid \mathbf{y}_{1:k-1}} \left[ \exp(\mathbf{r}_k^m) \mid \mathbf{x}_k \right] \overset{(d)}{=} \exp\left( \hat{\mathbf{r}}_{k|k-1}^m + \tfrac{1}{2} \mathbf{\Lambda}_{\hat{\mathbf{r}}_{mm}} \right)
\end{aligned}
$$

where in (a) we simplify using $\mathbf{y}_k$'s conditional independence from the past $\mathbf{y}_{1:k-1}$, in (b) we simplify based on $\mathbf{y}_k \mid \mathbf{x}_k \sim \text{Poisson}(\exp(\mathbf{r}_k))$, in (c) we use the Taylor series approximation of $\exp(\exp(\mathbf{r}_k))$ for small $\exp(\mathbf{r}_k)$, and (d) is simply the mean of a log-normal random variable. Note that $\hat{\mathbf{r}}_k = \boldsymbol{C}_{\mathbf{r}} \hat{\mathbf{x}}_{k|k-1} + \boldsymbol{b}$ and $\mathbf{\Lambda}_{\hat{\mathbf{r}}} = \boldsymbol{C}_{\mathbf{r}} \mathbf{\Lambda}_{\hat{\mathbf{x}}_{k|k-1}} \boldsymbol{C}_{\mathbf{r}}^T$, where $\hat{\mathbf{x}}_{k|k-1}$ is the current estimate for the state and $\mathbf{\Lambda}_{\hat{\mathbf{x}}_{k|k-1}}$ the state-prediction covariance (appendix A.4).

### A.6 POSSIBILITY OF LEARNING UNSTABLE MODES IN SMALL DATA REGIMES

Subspace identification methods generally only converge to the correct system parameters asymptotically (see figure 1a-b), as the empirically estimated covariances also converge to their true values (Van Overschee & De Moor, 1996). For finite samples, however, there will always be some error in the learned parameters. Although such errors are generally benign, extreme scenarios can result in unstable state dynamics, i.e., the identified $\boldsymbol{A}$ has at least one eigenvalue with magnitude larger than 1. In simulations (figure 1), we excluded learned models that were unstable from the reported mean performances, reflecting in the reduced number of samples in the standard error of the mean (s.e.m). For training set sizes typical of neuroscience datasets, the occurrences of unstable models was rare, with only 2 unstable systems for 1e5 training samples and no unstable systems for 1e6 training samples.

### A.7 COMPUTATION TIME DETAILS

We measured the learning time of our method and, additionally, the inference time associated with using a point-process filter for state estimation. Using one session of NHP data (section 4.2), we repeatedly trained on 25 distinct time-series datasets. Each dataset consisted of a 6097-by-15 matrix (timesteps-by-features) of Poisson observations and a 6097-by-4 matrix of Gaussian observations. The training time of our algorithm averaged across 25 trials was 0.33s (including 0.072s spent on the convex optimization problem outlined in 3.2.3). Further, we also measured the inference time for our 1524-sample testset to be 0.33s (i.e., approximately 0.2ms per 50ms timestep).

Most of the computational cost of our algorithm is involved in the matrix operations associated with 1) computing the necessary covariance/Hankel matrices, 2) performing the moment conversion, and 3) performing the SVD of the future-past Hankel matrices. To perform the moment conversion our method requires a covariance matrix for stacked future-past Poisson-Poisson observations (section 2.1) and a future-past Gaussian-Poisson Hankel matrix (section 3.2.1). Both of these empirical estimates of second-order covariances are computed using matrix multiplications which scale with the number of samples. As an example, we can consider the setup used for the computational cost analysis here, wherein $n_y = 15$ and $i_y = 10$ (horizon). The computed square Poisson-Poisson covariance matrix was of dimension $2 * n_y * i_y = 2 * 10 * 15$ and was the result of a matrix multiplication between two matrices of dimension (2*10*15)-by-6078, where $6078 = \text{timesteps} - 2 * i_y + 1$. Thus, this operation would scale linearly with the length of the training data. Similarly, the computational cost of this matrix multiplication scales linearly with feature dimension and horizon. The remaining operations (i.e., the SVD and the moment conversion itself) are functions of the latent-state dimension and the feature dimensions for each observation timeseries.

## A.8 Comparison against Laplace-EM PLDS

As further comparison with existing learning algorithms for Poisson generalized-linear dynamical systems, we also performed a preliminary comparison with Laplace-EM. We used the same NHP data as in section 4.2 but limited our analysis to the first session from the manuscript (i.e., 20160915/01). Preprocessing and other analysis details were as described in section 4.2. We implemented Laplace-EM using a well-known publicly available library for state-space modeling via EM (`https://github.com/lindermanlab/ssm`). We used the default settings for Laplace-EM from the above library and ran the optimization for 100 iterations. We used a state dimension of $n_x = 8$ for both PG-LDS-ID and Laplace-EM, and for our method we extracted all latent states using the first stage (i.e., $n_1 = n_x = 8$); the dimension was selected based on the results in Fig. 2. Finally, for our decoding comparison with EM we used an approach similar to our PLDSID analysis, wherein we model the Poisson neural dynamics first with EM, estimate the latent states using the learned model, and finally regress the latent states to the second observation time-series, i.e., the Gaussian behavior (as described in appendix A.4). The results are presented in Table 1:

Table 1: Results for Laplace-EM comparison

| Method | Latent size | Gaussian prediction CC | Poisson self-prediction AUC |
|---|---|---|---|
| **PG-LDS-ID** | $n_1 = n_x = 8$ | $0.4720 \pm 0.0097$ | $0.6533 \pm 0.0018$ |
| **Laplace-EM** | $n_x = 8$ | $0.3884 \pm 0.0120$ | $0.6718 \pm 0.0022$ |

We find that PG-LDS-ID outperforms Laplace-EM in decoding behavior from neural activity (i.e., predicting the Gaussian observations). This is due to PG-LDS-ID's ability to dissociate shared Poisson-Gaussian dynamics and prioritize their identification, whereas Laplace-EM is optimized on Poisson log-likelihood only. Further, PG-LDS-ID's resulting model achieves a slightly lower Poisson self-prediction AUC compared to Laplace-EM, which is unsurprising due to the use of the first stage only. As demonstrated in figure 2, the optional second stage of our method can additionally be used to learn any remaining Poisson dynamics and match Laplace-EM's self-prediction AUC.

## A.9 Second Biological Dataset Results

We performed a less comprehensive validation of our method on a second (independent) biological public dataset from the Miller lab with a different behavioral task (Lawlor et al., 2018; Perich et al., 2018). The task for this dataset involved a NHP controlling a cursor via a manipulandum to reach random targets on the screen sequentially. We used the same preprocessing, inner cross-validation, and modeling procedures as described in section 4.2. For both methods we used a latent-state dimension of $n_x = 8$. For PG-LDS-ID we used only stage 1 (i.e., $n_1 = n_x = 8$). We performed the analysis on only one session of the data, using random subsets of 15 single-unit channels (similar to the analysis in section 4.2). Results (Table 2) are similar to those for the first dataset in 2. We find that our method outperforms PLDSID in terms of behavior decoding. Also, as expected, the neural self-prediction at this dimension is lower than PLDSID, due to the prioritization of shared dynamics (i.e., the use of stage 1 only). However, as demonstrated in 2, we could add the second stage with enough latent state dimensions such that PG-LDS-ID's neural self-prediction improves.

Table 2: Results for the second NHP dataset

| Method | Latent size | Gaussian prediction CC | Poisson self-prediction AUC |
|---|---|---|---|
| **PG-LDS-ID** | $n_1 = n_x = 8$ | $0.4025 \pm 0.0133$ | $0.6176 \pm 0.0054$ |
| **PLDSID** | $n_x = 8$ | $0.3415 \pm 0.0114$ | $0.6569 \pm 0.0068$ |

