# OpenReview forum: "Spectral learning of shared dynamics between generalized-linear processes"
_ICLR.cc/2024/Conference — Submitted to ICLR 2024_

### Official Review · Reviewer_XxLi · 2023-10-31

**Soundness:** 2 fair
**Presentation:** 2 fair
**Contribution:** 2 fair
**Rating:** 6
**Confidence:** 1

**Summary:**

I am unable to assess this paper and have alerted the ACs to seek an opinion from different reviewers.

**Strengths:**

I am unable to assess this paper and have alerted the ACs to seek an opinion from different reviewers.

**Weaknesses:**

I am unable to assess this paper and have alerted the ACs to seek an opinion from different reviewers.

**Questions:**

I am unable to assess this paper and have alerted the ACs to seek an opinion from different reviewers.

---

### Official Review · Reviewer_NhF4 · 2023-11-01

**Soundness:** 3 good
**Presentation:** 2 fair
**Contribution:** 3 good
**Rating:** 8
**Confidence:** 4

**Summary:**

This paper proposed a new method called PG-LDS with its corresponding inference and learning algorithm that can find both shared and residual dynamics from coupled observations. Experiments on both simulated and real-world dataset validate the effectiveness of the proposed model and corresponding algorithms. The provided algorithm can be generalized to other similar models with paired observationa and shared latent dynamics.

**Strengths:**

* The whole paper is clear in presentation. Method derivation is detailed with maths.
* The new model is interesting to me, since the usual way of treating the behavior data of a neural dataset is to treat it as external input, or some ground truth to be compared. This paper provides a new perspective of dealing this problem. Specifically, the Poisson spike train and behaivor data are treated as a coupled dataset with shared latent dynamics. By this way, we are able to use both the spike train data and the behavior data to find some common factors that accounts for the observations in an experiment.

**Weaknesses:**

* First line of sec 2.2, typo: "Given an $H$". Page 3 -2 line, typo: "we need to".
* See questions.

**Questions:**

* What is "either colored or white" in Sec 3.1. This is confusing.
* In Eq. 5, why Gaussian observations $z_k$ don't include a bias term? What's the distribution of $\epsilon_k$?
* What about the comparisons of the log-likelihood on test datasets?
* Are there any existing models or methods that can learn shared latent dynamics from the joint dataset: e.g. neural spike trains plus movement?
* Have authors tried other datasets? Is the proposed model widely applicable to similar tasks?
* Since authors claim that the algorithm is able to be genearalized to non-Poisson/non-Gaussian model, have authors tried that on at least some synthetic datasets from simple models, which are not Poisson+Gaussian?

---

> ### Author Response · Authors · 2023-11-21
>
> We thank the reviewer for taking the time to review our manuscript and provide feedback. We address each of the raised points inline below.
>
> > What is "either colored or white" in Sec 3.1. This is confusing.
>
> White noise refers to temporally uncorrelated zero-mean Gaussian noise whereas colored noise refers to temporally correlated zero-mean noise. We have made the following clarification in the manuscript:
>
> *“(either white, i.e., zero-mean temporally uncorrelated Gaussian noise, or colored, i.e., zero-mean temporally correlated Gaussian noise)”*
>
> We had also provided more context on how colored noise is simulated in appendix section A.3.1:
>
> *“For every simulated model, the colored noise for the Gaussian process, $\epsilon_k$, was taken as the output of a 4-dimensional latent linear dynamical system, with random parameters that were generated similarly. By using a general colored noise, we can simulate dynamics present in the continuous modality that are unshared with the discrete modality.”*
>
> > In Eq. 5, why Gaussian observations $z_k$ don't include a bias term? What's the distribution of $\epsilon_k$?
>
> Because $z_k$ is a linear Gaussian process with stationarity assumptions, a bias term would effectively correspond to an affine shift. As a result, and without loss of generality, we can assume the random process is zero-mean (i.e., no bias term). All predictions can be done with demeaned time-series and, if desired, the constant mean can be added back once inference is completed. For real data, a bias term can be learned by computing a mean parameter across time for the training data stream and demeaning the data as a preprocessing step before fitting the model. The learned bias can then be added back to the predicted time series for evaluation.
>
> We refer the reviewer to the previous response in regards to the distribution of $\epsilon_k$:
>
> *“(either white, i.e., zero-mean temporally uncorrelated Gaussian noise, or colored, i.e., zero-mean temporally correlated Gaussian noise)”*
>
> > What about the comparisons of the log-likelihood on test datasets?
>
> We thank the reviewer for the suggestion. We are working on adding log-likelihood as another metric in the test datasets.
>
> > Are there any existing models or methods that can learn shared latent dynamics from the joint dataset: e.g. neural spike trains plus movement?
>
> We thank the reviewer for their question. No analytical methods exist for learning shared latent dynamics from joint Poisson and Gaussian datasets, including for generalized linear dynamical system models which is what we focus on here.
>
> It is worth noting that there do exist nonlinear deep learning methods for fitting multiple time-series, as cited in the manuscript. However, our work focuses on generalized-linear dynamical systems, which have remained popular given their interpretability for neuroscience investigations, analytical properties, rich theory, and broad applicability for developing real-time brain-computer interfaces. Thus, our goal here is to enable such generalized-linear dynamical systems models to prioritize the learning of shared dynamics between Poisson and Gaussian time-series data and dissociate them from disjoint dynamics.
>
> > Have authors tried other datasets? Is the proposed model widely applicable to similar tasks?
>
> We have now added a completely new dataset and show that the same results/conclusions hold. This is in a new appendix section. The new dataset is an independent public NHP dataset (CRCNS pmd-1 dataset from the Miller lab), with a different behavioral task involving moving a cursor to random targets. Briefly, we compared stage 1 of PG-LDS-ID against PLDSID when both methods have $n_x=8$ latent states. As in dataset 1 (Fig. 2), our method outperformed PLDSID in behavior decoding. Further, the neural self-prediction at this dimension is lower than PLDSID, as expected, due to the use of stage 1 only. The cross-validated results were
>
> |Method|Poisson self-prediction AUC (mean&plusmn;STE)|Gaussian prediction CC (mean&plusmn;STE)|
> |-|-|-|
> |PG-LDS-ID|0.6176&plusmn;0.0054|**0.4025 &plusmn;0.0133**|
> |PLDSID|**0.6569&plusmn;0.0068**|0.3415 &plusmn;0.0114|

---

> > ### Author Response · Authors · 2023-11-21
> >
> > > Since authors claim that the algorithm is able to be generalized to non-Poisson/non-Gaussian model, have authors tried that on at least some synthetic datasets from simple models, which are not Poisson+Gaussian?
> >
> > We thank the reviewer for their question. Generalization to other distributions requires changes only in the initial “moment transformation” step of the algorithm, and the remaining steps remain applicable. While extending this moment transformation step to other distributions is largely straightforward, it still involves some derivation and validation. For example, the focus of the recent Stone et al. paper was to extend PLDSID from Poisson to Bernoulli observations. Thus, we have left this extension for future work. As an example, an extension to our method could change the moment transformation step to derive it for Bernoulli observations (e.g., not unlike how Stone et al. extend the unimodal PLDSID to Bernoulli observations) to enable multimodal modeling of Bernoulli-Gaussian or Bernoulli-Poisson data. Other key steps from our work, including the two-stage learning, remain applicable regardless of data distributions.
> >
> > References:
> > Iris R Stone, Yotam Sagiv, Il Memming Park, and Jonathan W. Pillow. Spectral learning of bernoulli linear dynamical systems models for decision-making. Transactions on Machine Learning Research, 2023. ISSN 2835-8856
> >
> > > First line of sec 2.2, typo: "Given an H". Page 3 -2 line, typo: "we need to".
> >
> > We thank the reviewer for their comment and have now revised these issues.

---

> > ### Comment · Reviewer_NhF4 · 2023-11-22
> >
> > Thanks for the authors' detailed response. I'm more confident on this paper, but I still think test (predictive) likelihood is an important metric to be added. Even though it might not be the most effective one in this case since there are two or multiple types of observations, test likelihood is still the most widely applied (or even the only one that can always be applied on real-world datasets). I understand running extra experiments requires some time. Given such a situation, I would like to raise my confidence to 4 and raise my score to 8. However, I think a score of 7 is more suitable, but this option is not available here. Overall, I think this paper is good. Thanks again!

---

### Official Review · Reviewer_vavJ · 2023-11-01

**Soundness:** 3 good
**Presentation:** 2 fair
**Contribution:** 3 good
**Rating:** 8
**Confidence:** 2

**Summary:**

The paper presents a novel analytical approach, the PG-LDS-ID algorithm, designed for modeling Poisson data streams while disentangling shared dynamics with Gaussian data streams. This capability addresses the challenge of predicting the dynamics of one data stream from another with different statistical properties. Through simulations and real-world data, the authors demonstrate the effectiveness of their method in accurately identifying shared Poisson dynamics with Gaussian observations. The proposed algorithm's flexibility extends to various generalized-linear models, making it a valuable tool for modeling shared and distinct dynamics in data streams across diverse application domains.

**Strengths:**

The paper's most notable strength lies in its innovative decomposition technique introduced through Equation (6). This decomposition significantly simplifies the modeling of shared dynamics between data streams with different statistical properties. By breaking down the problem into manageable components, the paper enhances the overall approach's ease of handling and implementation.

For practical applicability, the introduced decomposition technique, as demonstrated in the paper, holds practical applicability in real-world scenarios. By simplifying the modeling of shared dynamics, the method provides a valuable tool for researchers in different domains. This practical aspect strengthens the paper's significance as it offers a solution that can be directly applied to address challenging problems.

**Weaknesses:**

1. While the decomposition introduced in Equation (6) is a notable strength, it lacks clarity regarding the conditions under which it can be effectively implemented. The paper does not sufficiently discuss the scenarios where this decomposition may not be feasible, and whether alternative methods should be considered.

2.  After the system decomposition, it is evident that r depends on both x^1 and x^2. However, the paper does not sufficiently explain why, in Section 3.2.1, C_r^1 can be estimated independently without considering C_r^2.

3. The experimental evaluation in the paper primarily compares the proposed algorithm with PLDSID, which was introduced in 2012. It is essential to explore whether newer and more competitive algorithms have been developed. A more comprehensive comparative analysis involving the most up-to-date methods would provide a clearer picture of the proposed algorithm's strengths and weaknesses in the current research landscape.

**Questions:**

See Weaknesses.

---

> ### Author Response · Authors · 2023-11-17
>
> We thank the reviewer for taking the time to review our manuscript and provide feedback.
>
> > While the decomposition introduced in Equation (6) is a notable strength, it lacks clarity regarding the conditions under which it can be effectively implemented. The paper does not sufficiently discuss the scenarios where this decomposition may not be feasible, and whether alternative methods should be considered.
>
> We thank the reviewer for their feedback. One can prove that any linear dynamical system can be written in this block-structured form by applying a change of basis (a similarity transform), and so this block-structured form does not lose generality (Katayama, 2006). We have now added a new appendix section in the manuscript that clarifies the mathematical derivation of the block structure in equation (6). Briefly, based on linear systems theory an equivalent basis for the latent space always exists that has the block structure in equation (6), in which the latent subspace that is observable via $z_k$ is placed as the first few dimensions of the latent state (see for example Theorem 3.8 from Katayama, 2006). Importantly, the blocked formulation from equation (6) also covers the special cases of non-blocked formulation when $n_1=n_x$, i.e., when all latent states contribute to both data streams (equivalent to full-rank observability). For more details about why this formulation is general and how the appropriate value for $n_1$ can be determined in real data, we invite the reviewer to see our longer response to reviewer QEhH, who had a similar question.
>
> References:
> Tohru Katayama. Subspace Methods for System Identification. Springer London, 2005. doi: 10.1007/1-84628-158-x.
>
> > After the system decomposition, it is evident that $r$ depends on both $x^1$ and $x^2$. However, the paper does not sufficiently explain why, in Section 3.2.1, $C_r^1$ can be estimated independently without considering $C_r^2$.
>
> This is a great question. We have made edits to section 3.2.1 and appendix A.1.2 to make this point more clear. The core reason for why $C_r^{(1)}$ can be estimated in the first stage of our algorithm without the need to simultaneously estimate $C_r^{(2)}$ is the block structure in equation (6), which is obtained without loss of generality, as noted in the previous question. Since the top right block of A in equation (6) is zero, we can see that the evolution of the shared dynamics (captured by $x^{(1)}$) does not depend on the evolution of the distinct dynamics in the predictor data stream (captured by $x^{(2)}$). This allows the parameters associated with the shared dynamics to be learned directly, as derived in the first stage of our method, and for the remaining parameters ($C_r^{(2)}$) to be learned in an optional second stage (section 3.2.2). We present the detailed derivations in appendix A.1.2, specifically equations (20), (21), and (22). Briefly, the block structure introduced in equation (6) yields a block structure to the observability and controllability matrices defined in equation (16) as $\mathbf{H} = \Gamma\Delta$. Further, this block structure can be used in conjunction with SVD to separate $H$ into two components defined as $H = \Gamma_{\mathbf{r}}^{(1)}\Delta^{(1)} + \Gamma_{\mathbf{r}}^{(2)}\Delta^{(2)}$, with $\Delta^{(1)}$ orthogonal to $\Delta^{(2)}$. As a result, right multiplication with the pseudoinverse of $\Delta^{(1)}$ allows independent estimation of $C_r^{(1)}$.

---

> > ### Author Response · Authors · 2023-11-17
> >
> > > The experimental evaluation in the paper primarily compares the proposed algorithm with PLDSID, which was introduced in 2012. It is essential to explore whether newer and more competitive algorithms have been developed. A more comprehensive comparative analysis involving the most up-to-date methods would provide a clearer picture of the proposed algorithm's strengths and weaknesses in the current research landscape.
> >
> > We thank the reviewer for this comment. We would like to clarify two points. First, because the method presented in this work is an analytical method for generalized linear dynamical systems, we chose to baseline against existing analytical algorithms within the same model class. Thus, the most appropriate baseline was PLDSID, which is the primary analytical approach for modeling linear dynamical systems with Poisson observations, and a special case of our algorithm (when only the second stage is used). Second, we note that beyond analytical methods for linear dynamical systems, we have also performed a comparison against a numerical Laplace-EM PLDS method, the results of which are presented in appendix A.6 of the original submission. Expectation maximization (EM) is a popular approach for unsupervised training of linear dynamical models with Poisson observations; EM iteratively optimizes the Poisson log-likelihood and thus benefits from the optimality properties of maximum likelihood estimation in general. Our comparisons against Laplace-EM again show that models learned using stage 1 of our algorithm more accurately learn the shared dynamics, yielding a higher decoding accuracy. This improved decoding performance is achieved despite our method being computationally much more efficient by being analytical and non-iterative. Finally, of note, deep learning methods can also be used to fit nonlinear models for neuroscience data, as cited in the manuscript. However, our work focuses on generalized-linear dynamical systems, which have remained popular given their interpretability for neuroscience investigations, analytical properties, rich theory, and broad applicability for developing real-time brain-computer interfaces. Thus, our goal here is to expand the capabilities of linear dynamical models, given their wide-spread use in neuroscience and engineering. In particular, we now allow these models to tease apart and accurately identify shared dynamics between Poisson and Gaussian time-series data.

---

> > ### Comment · Reviewer_vavJ · 2023-11-20
> >
> > For the second question, I would like further understand why C_r^1 can be independently estimated. For now, my understanding is that in stage 1, the variable r is only involved in the computation of its covariance with z, and the contribution of C_r^2 to the covariance is zero. Therefore, C_r^1 can be independently estimated. Is this understanding correct?

---

> > > ### Author Response · Authors · 2023-11-21
> > >
> > > Thank you for the follow-up. Yes, your understanding is correct. To show that this is the case we can consider the parameter expansion of the lag terms $\boldsymbol{\Lambda}\_{zr_\tau} = \text{Cov}(z_{k+\tau}, r_k)$ that make up the cross-covariance Hankel matrix $\boldsymbol{H}\_{zr} = \text{Cov}(z_f, r_p)$, as defined in appendix section A.1.2. For example, expanding the $\tau = 1$ lag term:
> > >
> > > $\text{Cov}(z_{k+1}, r_k) = \text{Cov}(C_z x_{k+1}, C_r x_k) =  \text{Cov}(C_z A x_{k}, C_r x_k) = C_z A \Sigma_{x} C_r^T = \begin{bmatrix}C_z^{(1)} & \mathbf{0} \end{bmatrix} \begin{bmatrix}A_{11} & \mathbf{0} \\\\ A_{21} & A_{22} \end{bmatrix} \Sigma_{x} \begin{bmatrix} C_r^{(1)T} \\\\ C_r^{(2)T}  \end{bmatrix} =$
> > > $= \begin{bmatrix} C_z^{(1)} A_{11} & \mathbf{0} \end{bmatrix} \begin{bmatrix}\Sigma_{x}  C_r^{(1)T} \\\\ \Sigma_{x}  C_r^{(2)T}  \end{bmatrix} = C_z^{(1)} A_{11} \Sigma_{x}  C_r^{(1)T}  $
> > >
> > > where $\Sigma_{x}$ is the covariance of the latent states. For the other lags $\tau$ we can show similar expansions. Thus, due to the block structure, the cross-covariance Hankel matrix $\boldsymbol{H}_{zr}$ has no contribution from $C_r^{(2)}$.

---

> > > > ### Comment · Reviewer_vavJ · 2023-11-23
> > > >
> > > > Thank author for the detailed reply. Considering the discussion, I raise my score to 8.

---

### Official Review · Reviewer_1pgP · 2023-11-05

**Soundness:** 3 good
**Presentation:** 2 fair
**Contribution:** 2 fair
**Rating:** 6
**Confidence:** 3

**Summary:**

The authors present a method for estimating latent dynamical processes observed through two distinct observations processes - one that delivers continuous-time observations (here Gaussian observation process) and one that produces discrete time observations (here Poisson process). This in a practical setting might be continuous-time behavioral trajectories (i.e. arm movements) and neuronal activity data recorded from relevant brain regions.
The authors follow a covariance-based subspace system identification method to estimate what they call both the shared dynamics observed through the Gaussian and Poisson observation processes, and the disjoint (residual) dynamics that are not observable by both processes, but only through the Poisson observation process.

The proposed method follows a two stage approach for learning first the shared dynamics that are jointly observed by the two observation processes, and a second stage that identifies the residual dynamics only observed by the Poisson process.
They demonstrate their method on a simulated linear model system, and on non-human primate dataset of discrete population spiking activity recorded during continuous arm movements.

**Strengths:**

- Under the assumption of linearity the authors can estimate the dimensionality of the latent process, and of the shared and residual subspace between the two observation processes.
- An issue with covariance-based sub-space identification methods is that there is no guarantee that a valid set of parameters that satisfy the positive semi-definite covariance sequence will be recovered. Here the authors optimise by ensuring the validity of noise statistics.

**Weaknesses:**

- The main weakness is the strong assumption of latent linear dynamics, that nevertheless is essential for the development of the method. However for the applications the authors have in mind, most systems are nonlinear. Thus I would expect some comparison of the performance of the method when applied to observations generated by a latent nonlinear system (as also mentioned in the questions below).
- The authors do not outline the difference of the proposed approach with recent similar approaches that use subspace identification method relying both on behavioural and neural data, i.e. [1], [2].

**Questions:**

- The dimensionality of the shared and residual dynamics can be estimated from the singular value decomposition/low rank approximation of the Henkel matrices, as mentioned in Appendix A.1.5. However this estimation method will be accurate under the assumption that the observed latent dynamical system is indeed linear (instead of approximated by a linear system). Do the authors have any estimation on how the method will perform in cases where the latent system is nonlinear? In this case both the evolution equation of the latent process will be an approximation, but also importantly the dimensionalities of the shared and residual dynamics will probably be estimated inaccurately. I think it would be helpful to see systematic evaluations on how the method performs:
   - i) when the dimensionalities of the subspaces are misestimated, and
  - ii) when the linearity assumption does not hold (assuming that the dimensionalities of the subspaces have been correctly assessed). For example In Figure 2 (where the method is applied on primate data) I wouldn’t say that the estimation of the dimensionality actually works (Figure 2 a and b).
- In Section 3.1 in mentioning the model parameters that the method identifies, the authors do not include the noise variance of the Gaussian observation process $z_k$ (I.e. the characteristics of the noise term $epsilon_k$). Similarly in Section 3.2.3.
   - Does this mean that the noise variance of the Gaussian observation process does not influence the performance of the method. Can the authors comment on this?
   - Moreover I would also include the dimensionality of the latent process and the dimensionality of the shared subspace dynamics in the model parameters that are estimated.
- How would the approach compare to estimation based on the Gaussian observation process? In Figure 1 the authors compare their framework with the PLDSID one, but I wonder how the method would compare to a method relying only on the continuous observations for identifying the shared subspace dynamics.
- In Figure 1 caption the authors mention: “PG-LDS-ID stage 1 used a dimensionality given by $\min(4, n_x )$ “. Can you explain what is meant with this phrase, and how the value 4 was chosen?

Minor comments on writing:

- In the first paragraph of the introduction the authors mention “Second, disjoint dynamics present in either observation can obscure and confound modeling of their shared dynamics”. Up to here in the text it is still unclear what “disjoint dynamics” is, and what “each observation” refers to. For the latter I would propose to replace with “each observation stream” or something along these lines. For the first, I would use something referring to “uncorrelated” part of the dynamics or residual as you call it later, but with a brief explanation what exactly is meant with this term.

- Similarly in Page 4 point 2., the authors mention the transition matrix presented later in the text without giving more detail at this point transition matrix of which of the processes they consider they refer to.
- In the introduction the authors refer to the part of the dynamics that is only observable through one observation process as “disjoint” part, while in the main text they refer to this part as residual dynamics. I would propose to stick to one of those terms (preferably the latter one) to avoid confusing the readers.

----

**References:**

[1] Ahmadipour, P., Sani, O. G., Pesaran, B., & Shanechi, M. M. (2023). Multimodal subspace identification for modeling discrete-continuous spiking and field potential population activity. bioRxiv, 2023-05.
[2] Vahidi, P., Sani, O. G., & Shanechi, M. M. (2023). Modeling and dissociation of intrinsic and input-driven neural population dynamics underlying behavior. bioRxiv, 2023-03.

---

> ### Author Response · Authors · 2023-11-22
>
> > The authors do not outline the difference of the proposed approach with recent similar approaches that use subspace identification method relying both on behavioural and neural data, i.e. [1], [2].
>
> We have expanded our discussion of the differences with these works. Briefly, neither of these methods can dissociate or prioritize the shared dynamics between Poisson and Gaussian data streams as we do here, nor do they aim to predict one data stream from the other. The first cited reference models the collective dynamics (i.e., union) of concatenated spiking and field potential signals, not their shared dynamics. This work also does not dissociate shared versus distinct dynamics into separate latent states, and does not prioritize learning of the shared dynamics – both of which are our goals. Depending on data statistics, such a multimodal approach may dedicate the latent states largely to explaining the modality with the higher variance and/or to the mixture/union of dynamics (i.e., both shared and residual). Thus, it may fail to learn some shared dynamics that can be masked by the residual dynamics. In contrast, our approach explicitly learns the shared dynamics first (with priority) by forming a Hankel matrix with the secondary data stream as the future and the predictor data stream as the past. We make a note of this point in Introduction section (end of the first paragraph on the second page):
>
> *“Finally, prior multimodal learning algorithms do not explicitly tease apart the shared vs. disjoint dynamics in a predictor (primary) time-series, but instead model the collective dynamics of two modalities in the same latent states (Abbaspourazad et al., 2021; Kramer et al., 2022; _Ahmadipour et al., 2023_).”*
>
> Regarding the second cited reference, there are three main distinctions. First, the cited work is just for linear Gaussian processes and therefore is not applicable to Poisson or other generalized-linear processes considered here. Second, the subspace identification approach developed in the cited work uses projection-based instead of covariance-based subspace identification methods, which is not applicable to Poisson. Finally, instead of modeling Poisson and Gaussian shared dynamics, which is our goal here, the focus of that work is to incorporate a third data stream (i.e., input) into the model to dissociate input-driven and intrinsic dynamics (i.e., model Gaussian time-series in the presence of input). We have added a discussion of these differences to the manuscript.
>
> > In Section 3.1 in mentioning the model parameters that the method identifies, the authors do not include the noise variance of the Gaussian observation process z_k (i.e. the characteristics of the noise term epsilon_k). Similarly in Section 3.2.3. Does this mean that the noise variance of the Gaussian observation process does not influence the performance of the method. Can the authors comment on this?
>
> We thank the reviewer for their great question and clarify two points. First, the model parameters stated in section 3.1 correspond to the parameters required by the point-process filter for state estimation (i.e., inference) (Eden et al., 2004). However, if desired, the noise covariance term under Gaussian assumptions can be learned by computing the covariance of the prediction residuals as such $E[(\hat{z}_k - z_k)(\hat{z}_k - z_k)^T]$, where $\hat{z}_k$ denotes the predicted value of $z$ and the expectation denotes the empirical average across all time samples. When $\epsilon_k$ is not white, the behavior prediction residuals under Gaussian assumptions can be computed in the same way (i.e., $\hat{z}_k - z_k$) and modeled using Gaussian SID.
>
> The second point is in regards to the impact the Gaussian process’ noise has on learning. The effect is similar to that of the disjoint dynamics present in the primary (i.e., Poisson) data stream: it reduces the overall signal-to-noise ratio of the shared dynamics present in both observation data streams. However, with increasingly more training samples and improved estimates of the second-order moments (i.e., covariances), the algorithm would be more robust to the impact of the Gaussian observation noise.
>
> We have included a new appendix section discussing the effect of Gaussian observation noise on learning and how its statistics (i.e., $\epsilon_k$) can be learned if necessary.
>
> > In Figure 1 caption the authors mention: “PG-LDS-ID stage 1 used a dimensionality given by min(4, $n_x$)”. Can you explain what is meant with this phrase, and how the value 4 was chosen?
>
> For the simulation results presented in Figure 1c and d, the true model parameter for the shared latent states is $n_1 = 4$. For each $n_x$ value along the x-axis, the corresponding $n_1$ (i.e., the dimensionality of the first stage which aims to identify the shared latents) is set to be either the minimum of the true $n_1=4$ value or $n_x$, as $n_x$ designates the total number of latents ($n_1 + n_2$).

---

> > ### Author Response · Authors · 2023-11-22
> >
> > > The dimensionality of the shared and residual dynamics can be estimated from the singular value decomposition/low rank approximation of the Henkel matrices, as mentioned in Appendix A.1.5. However this estimation method will be accurate under the assumption that the observed latent dynamical system is indeed linear (instead of approximated by a linear system). Do the authors have any estimation on how the method will perform in cases where the latent system is nonlinear? In this case both the evolution equation of the latent process will be an approximation, but also importantly the dimensionalities of the shared and residual dynamics will probably be estimated inaccurately.
> >
> > We thank the reviewer for their comment and provide clarifications. First, we would like to clarify that beyond simulations, we have validated our method on real datasets that are in no way constrained to be linear. The dataset in Figure 2 consisted of spiking activity from populations of neurons and of continuous arm kinematics. The fact that the method performed well in this dataset shows the method’s utility to nonlinear neuroscience datasets. To further validate the model’s utility, we now also add a completely new and independent public NHP dataset (CRCNS pmd-1 dataset from the Miller lab), with a different behavioral task. In a new appendix section, we again show that the same results/conclusions hold, further demonstrating the method’s utility and generalizability to neuroscience data. Together, these results show that while our approach only provides a generalized linear approximation of the data, the is approximation is reasonable in neuroscience. This conclusion is also consistent with many works in neuroscience that have long used generalized linear approximations successfully for modeling neural and behavioral data – suggesting the general utility of these approximations. Generalized linear models are also popular due to their interpretability, data efficiency, and amenability to real-time decoding and control (important for brain-computer interfaces).
> >
> > Second, regarding the latent state dimension, we agree that if there are model mismatches, e.g., data is nonlinear but the model is linear, the estimation of latent state dimension will not reflect that of the true data/model. But we emphasize that this correct dimension estimation is actually not necessary for accurate modeling of data; indeed, it is likely a larger dimensionality will provide better fit to data than the true dimensionality as follows. Let’s take the example of data coming from a nonlinear model. If one aims to approximate this data with a linear model, it will actually be harmful to use the true dimensionality of the ground-truth nonlinear model because the linear model will need a much higher dimension to be able to provide a good approximation of the nonlinear data. Thus, one would actually want to pick a higher dimensionality than that of the true dimensionality. In such a case, as an alternative to determining the latent state by inspecting singular values, one can determine the latent state dimension based on a metric of fit to data within cross-validation. Briefly, one can sweep over various state dimensions, evaluate the fit to data for models at each dimension, and find the smallest dimensionality that reaches peak fit to the data – as was done for the NHP results in Figure 2 and in the second biological dataset appendix section. Please see the response to reviewer QEhH for a detailed explanation of this procedure.
> >
> > > Moreover I would also include the dimensionality of the latent process and the dimensionality of the shared subspace dynamics in the model parameters that are estimated.
> >
> > We thank the reviewer for their suggestion. We have made changes to sections 2.1, 3.1, 3.2.1, 3.2.2, and 3.2.3, accordingly, to include the shared ($n_1$), disjoint ($n_2$), and total ($n_x$) latent state dimensions in the model hyperparameters to be learned. We have expanded our description of how these state dimension hyperparameters can be determined by sweeping over candidate values or by inspecting singular values in a new appendix section (also see response to reviewer QEhH for a summary of how these state dimensions can be determined).

---

> ### Author Response · Authors · 2023-11-22
>
> > How would the approach compare to estimation based on the Gaussian observation process? In Figure 1 the authors compare their framework with the PLDSID one, but I wonder how the method would compare to a method relying only on the continuous observations for identifying the shared subspace dynamics.
>
> We thank the reviewer for their question. The same mathematical issues will arise in this case and thus the performance of using only the Gaussian observation process for identifying the shared dynamics subspace would face the same shortcomings as using the discrete observations only. This is because identification of the shared subspace is dependent on the ratio of the shared vs. residual/unshared dynamics present in the observation process being modeled. If the residual/unshared dynamics explain the majority of the variance in the continuous observations, then these residuals will mask/confound the shared dynamics during identification from continuous observations. Indeed, because the identification method will aim to explain as much variance in the continuous observations as possible, the identification of the residual dynamics will precede identification of the shared dynamics and occupy most of the latent state dimensions. As such, the shared dynamics will be missed or inaccurately learned. In contrast, our approach prioritizes the identification of shared dynamics -- even if they are a minority of the total variance -- by looking at both continuous and discrete observations together during learning. Thus our method learns these shared dynamics more accurately. Only if the majority of the Gaussian observation variance is due to the shared dynamics, then using the continuous observations alone will yield comparable results to our method. However, this is typically not the case in most neuroscience datasets because signals are complex and a multitude of sources contribute to their variance, not a single source.
>
> > In the first paragraph of the introduction the authors mention “Second, disjoint dynamics present in either observation can obscure and confound modeling of their shared dynamics”. Up to here in the text it is still unclear what “disjoint dynamics” is, and what “each observation” refers to. For the latter I would propose to replace with “each observation stream” or something along these lines. For the first, I would use something referring to “uncorrelated” part of the dynamics or residual as you call it later, but with a brief explanation what exactly is meant with this term.
>
> We thank the reviewer for their suggestion. We have made the following adjustment to the sentence in question:
>
> *Second, residual (i.e., unshared or unique) dynamics present in each observation stream can obscure and confound modeling of their shared dynamics*
>
> > Similarly in Page 4 point 2., the authors mention the transition matrix presented later in the text without giving more detail at this point transition matrix of which of the processes they consider they refer to.
>
> We thank the reviewer for their comment. We have now revised the sentence in question to read as follows:
>
> *As a result, we derived a new least-squares problem for learning the components of the state transition matrix $\boldsymbol{A}$ corresponding to the unique predictor process dynamics (section 3.2.2), without changing the shared components learned in the first stage (section 3.2.1).*
>
> > In the introduction the authors refer to the part of the dynamics that is only observable through one observation process as “disjoint” part, while in the main text they refer to this part as residual dynamics. I would propose to stick to one of those terms (preferably the latter one) to avoid confusing the readers.
>
> We have revised the text to consistently use “residual” instead of “disjoint” throughout the text and define it upon first use, as noted above.

---

### Official Review · Reviewer_QEhH · 2023-11-05

**Soundness:** 3 good
**Presentation:** 4 excellent
**Contribution:** 3 good
**Rating:** 6
**Confidence:** 3

**Summary:**

This paper proposed a multi-stage algorithm based on method of moments and positive semidefinite programming to estimate a dynamical system, where the observations come from two processes with shared dynamics. The authors consider specifically the setting where one process is Gaussian and another is Poisson, and the latent state of the Gaussian process leads to better prediction of the Poisson process but not vice versa. Simulation results show that the proposed method can accurately recover the shared dynamic, and real data experiment shows that the proposed method has better prediction accuracies compared with prior work.

**Strengths:**

1. The paper provides an algorithm that is able to estimate the shared dynamics of two generalized linear process. Compared to previous work (Buesing et al 2012), the inclusion of another correlated process leads to better prediction.

2. By using second order moments, the proposed method can now deal with generalized linear processes instead of Gaussian.

**Weaknesses:**

1. The paper seems to be motivated by solid applications in neuroscience; but I am not sure if this is of interest to the more general machine learning community.

2. I am a little concerned about the structure of the dynamical system formulation. See questions.

**Questions:**

Why can you assume that the coefficients follow the block structure as in equation (6)? Is this something motivated by the application? If the true coefficient matrix $\mathbf A$ is non-zero on the upper right block, is the proposed method still going to work?

---

> ### Author Response · Authors · 2023-11-17
>
> We thank the reviewer for taking the time to review our manuscript and provide feedback. The major concern from the reviewer was in regards to the block structure of the dynamical systems formulation, specifically:
>
> > Why can you assume that the coefficients follow the block structure as in equation (6)? Is this something motivated by the application? If the true coefficient matrix is non-zero on the upper right block, is the proposed method still going to work?
>
> This is a great question. One can prove that any linear dynamical system can be written in this block-structured form by applying a change of basis (a similarity transform), and so this block-structured form does not lose generality (Katayama, 2006). We have now added a new appendix section in the manuscript that clarifies the mathematical derivation for the block structure in equation (6). Briefly, based on linear systems theory an equivalent basis for the latent space always exists that has the block structure in equation (6), in which the latent subspace that is observable via $z_t$ is placed as the first few dimensions of the latent state (see for example Theorem 3.8 from Katayama, 2006). This equivalent formulation, which is also known as the canonical Kalman form (for observability), is always available via an invertible linear transformation of the latent space, without any loss of generality. Formally, we define the dimension of the shared states (denoted by $n_1$) based on the rank of the observability matrix for the pair $(A, C_z)$. Importantly, this blocked formulation from equation (6) even covers the special case the reviewer refers to – when “the true coefficient matrix is non-zero on the upper right block” – which happens if $n_1=n_x$, i.e., when all latent states contribute to both data streams and the observability matrix is full-rank. In this case, only the first stage of learning would be applied as $n_1=n_x$ and so $n_2=0$. Importantly, though, within the application of modeling neural and behavioral data, we typically expect a minority of neural dynamics to be related to a particular behavior of interest and so we expect the most appropriate $n_1$ to be smaller than $n_x$ in the vast majority of cases, thus requiring both stages of learning.
>
> One can estimate the most appropriate $n_1$ and $n_x$ from the data as follows:
> 1. Sweep over values of $n_1$, increasing $n_1$ while keeping $n_x=n_1$ (i.e., using stage 1 only to learn). Quantify the prediction of the secondary data stream (e.g., behavior) in each case to find the $n_1$ at which the prediction plateaus or reaches a peak. This value gives the appropriate $n_1$. Alternatively, $n_1$ can be estimated as the number of non-zero (or non-negligible) singular values of the Hankel matrix $\boldsymbol{H}_{zr}$ (equations (8) and (9)).
> 2. Using the selected $n_1$ from above, sweep over values of $n_x$, starting from $n_1$ and increasing the latent state dimension. Quantify the self-prediction of the predictor data stream (e.g., spiking activity) in each case, and find the $n_x$ at which the self-prediction reaches a peak.
>
> We have also included discussion regarding the above procedure in the newly added appendix section in the manuscript.
>
> References:
> Tohru Katayama. Subspace Methods for System Identification. Springer London, 2005. doi: 10.1007/1-84628-158-x.

---

### Meta-Review · Area_Chair_iNM3 · 2023-12-07

**Metareview:**

The paper introduces a novel spectral method for modeling shared dynamics. The authors have effectively demonstrated its utility through simulations and real-world datasets. Notably, the approach offers a new perspective on treating neural and behavioral data, highlighting its ability to uncover shared latent dynamics. Despite the final positive review scores and effective responses from the authors, the decision leans towards rejection due to concerns about the paper's novelty, pointed out by several reviewers (and meta-reviewer). While the algorithm performs well in its designated tasks, its contributions do not appear to significantly advance the field in a novel direction. The core idea, while executed well, does not depart sufficiently from existing methods to warrant a higher score. The assumption of latent linear dynamics, though addressed, still limits the broader applicability and innovativeness of the approach. Additionally, the lack of comparison with more recent and possibly more competitive algorithms weakens the paper's position within the current research landscape. In conclusion, while the paper is technically sound and contributes to the field, its limited novelty and lack of significant advancement over existing methods are the primary reasons for recommending rejection.

**Justification For Why Not Higher Score:**

The core idea, while executed well, does not depart sufficiently from existing methods to warrant a higher score. The assumption of latent linear dynamics, though addressed, still limits the broader applicability and innovativeness of the approach. Additionally, the lack of comparison with more recent and possibly more competitive algorithms weakens the paper's position within the current research landscape.

**Justification For Why Not Lower Score:**

N/A

---

### Decision · Program_Chairs · 2024-01-16

Reject